# CAN TRANSFORMERS REALLY DO IT ALL?
# ON THE COMPATIBILITY OF INDUCTIVE
# BIASES ACROSS TASKS

**Damien Teney**[1,3]  **Liangze Jiang**[1,2]  **Hemanth Saratchandran**[3]  **Simon Lucey**[3]
[1]Idiap Research Institute  [2]EPFL  [3]Adelaide University

## ABSTRACT

Transformers are remarkably versatile and their design is largely consistent across a variety of applications. But are they optimal for any given task or dataset? The answer may be key for pushing AI beyond merely scaling current designs.

**Method.** We present a method to optimize a transformer architecture for a given dataset, which we use as a tool to study optimal task-specific inductive biases. This method replaces the most important non-linearities (GeLUs, softmax) with functions learned on held-out data. We then train the resulting architectures on other datasets, as a way to evaluate the compatibility between pairs of tasks.

**Findings.** On algorithmic toy tasks, we identify new architectures with dramatic improvements in learning speed, in- and out-of-distribution generalization, and stability across seeds. The new designs prove very task-specific however, and indicate that these tasks require inductive biases very different from those of standard transformers. On code and language modeling datasets, we also find architectures with consistent, yet smaller improvements. These designs transfer much better across datasets and domains (English & computer code).

**Implications.** Our results show that standard transformers are rarely a local optimum in the space of architectures. Simple alternatives can perform much better but sacrifice universality. This suggests that there may be room for improved architectures that better support multiple capabilities simultaneously, such as fluency and robust reasoning.

## 1 INTRODUCTION

**Inductive biases of transformers.** The recent history of machine learning has seen a convergence of architectures across modalities. Most state-of-the-art models for vision, language, and speech are based on transformers, barring only minor differences (Vaswani et al., 2017). This success contrasts with earlier task-specific models, and it has prompted the hypothesis that transformers implement very generic inductive biases[1] well suited to various types of real-world data (Goldblum et al., 2023; Teney et al., 2024; 2025). Considering the vast space of possible architectures, the following question remains (Q1).

> *Are transformers uniquely endowed with such generic inductive biases?*

**Uneven performance across domains.** Transformers perform remarkably, for example as language models. But they also fail on elementary tasks such as learning arithmetic (Nikankin et al., 2024). These failures have motivated a plethora of alternative components such as positional encodings (Cai et al., 2025; Jelassi et al., 2024) and attention mechanisms (Katharopoulos et al., 2020; Saratchandran et al., 2024b; Schlag et al., 2021). But these designs are rarely adopted beyond toy tasks. This suggests that the inductive biases of standard transformers may not be suited to domains as different as natural language and arithmetic. This raises the following question (Q2).

> *Should we even seek to address such different domains with the same learning method?*

---

[1]The *inductive biases* of a learning algorithm can be seen as a prior over the space of functions (Mitchell, 1980; Mingard et al., 2021) such that particular (types of) functions are favored among the many that fit the data. We focus on biases encoded in architectures, not in choices of optimizer, objective function, initialization, etc.

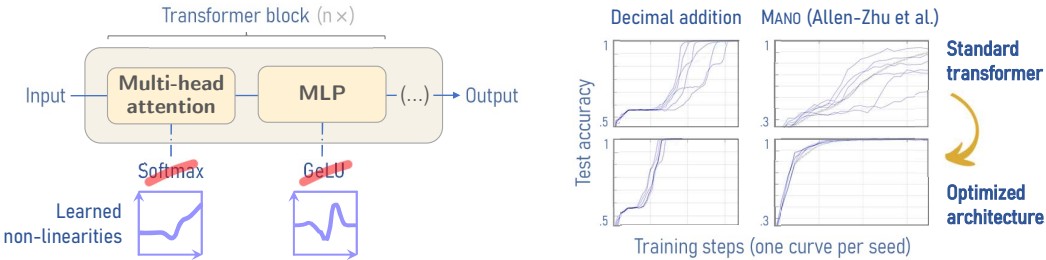

Figure 1: **Our approach to discover better task-specific inductive biases.** (Left) We replace the non-linearities in a transformer (softmax, GeLUs) with splines optimized from scratch for one chosen dataset. (Right) The new resulting architectures (with frozen splines as non-linearities) allow us to train models with dramatically better convergence, generalization, and stability across seeds, on algorithmic tasks and code/language modeling datasets. We also mix-and-match the new architectures across tasks (not pictured) to evaluate the compatibility of inductive biases across tasks.

The above questions are relevant for developing better AI systems. Current designs have largely relied on scaling up models and data (Mayilvahanan et al., 2025). This growth is not infinitely sustainable. and their deficiencies (e.g. in systematic generalization and data efficiency) indicate a need for better inductive biases. Understanding the suitability of the inductive biases of transformers to specific tasks (Q1) and the compatibility between tasks (Q2) are steps in this direction.

**Our approach.** We address the above questions with a method that seeks better inductive biases for a chosen dataset by tweaking the transformer architecture. We replace non-linearities (GeLUs, softmax) with parametrized splines, optimized on held-out data. This yields new architectures that match or surpass standard transformers. The improvement in learning speed and/or generalization indicates how far the standard transformer is from a local optimum in the space of architectures for a specific task (Q1). We also mix-and-match these new architectures across tasks to assess how the inductive biases tuned for one task perform for another, thus assessing their compatibility (Q2).

**Findings.** We study two domains: algorithmic skills and language modeling. For algorithmic skills, we use toy tasks commonly used to evaluate architectures (see e.g. Allen-Zhu 2025). For all considered tasks, our approach finds architectures that dramatically improve learning speed, generalization, and stability across random seeds (Section 3.1). Our task-specific variants of transformers are **vastly superior to standard designs, using only minor modifications** such as replacing the GeLUs. The cross-task evaluation also reveals that the new architectures are quite task-specific. This explains why many components proposed in the literature (e.g. attention mechanisms, positional encodings) are rarely adopted beyond toy tasks. These results also challenge the view that a single architecture can be optimal for a vast set of tasks (Goldblum et al., 2023).

For language modeling, we evaluate multiple datasets of natural language and computer code. In most cases, we find optimized architectures that slightly improve over a baseline transformer. We stress that these improvements are practically not directly useful, because standard components are more computationally efficient. But they matter indirectly, because they are evidence that **standard transformers are neither a unique nor a local optimum** in the space of architectures. In contrast to algorithmic tasks, the cross-task evaluation shows here that the improvements can transfer across natural language datasets, and across tokenization levels (character vs. subword). Overall, our results indicate that standard transformers are intrinsically better suited to modeling natural language than code, and clearly ill-equipped to learn algorithmic skills.

Our contributions are summarized as follows.

- **A method to optimize a transformer architecture** for any given dataset (Section 2). We replace GeLUs and softmaxes with parametrized components optimized on held-out data. The optimized architecture can then be used with standard training to evaluate its suitability to any other dataset.
- **An application to algorithmic tasks** (Section 3). We find that optimized architectures dramatically improve learning speed, generalization, and stability across seeds. They also prove very task-specific, showing the utility of inductive biases very different from standard transformers'.
- **An application to language modeling** (Section 4). We obtain small, albeit consistent improvements, showing that standard transformers are neither unique nor optimal designs, even for common code and natural language modeling tasks.

We discuss the implications for the development of future AI models in Section 6.

## 2 Proposed Method to Optimize Architectures

**Goal.** As a baseline architecture, we consider a standard decoder-only transformer (GPT-2-style, see details in Appendix B). Our goal is to evaluate whether this choice is optimal for specific datasets. We also seek better variants of transformers, i.e. identifying inductive biases better suited to each task. Evaluating the new architectures across tasks can then measure the compatibility of pairs of tasks. All the tasks we consider are formulated as sequence completion of natural language, computer code, or abstract tokens.

**Replacing non-linearities with learnable parametrized functions.** We replace the main non-linearities in a transformer with components that can be optimized (see Figure 1). Indeed, the main difference between a transformer and a simple linear model hinges on a few non-linear operations in the attention and MLP layers, which we will alter to obtain different inductive biases.

- An MLP layer is defined as: $x \leftarrow W' \phi(Wx + b) + b'$ where $x$ is a vector of activations, $W$, $W'$, $b$, $b'$ learned weights and biases, and $\phi : \mathbb{R} \to \mathbb{R}$ an element-wise non-linearity. In the baseline architecture, $\phi$ is a GeLU. In our model, $\phi_{\theta_{\mathrm{MLP}}}$ is a 1D linear spline parametrized by learnable keypoints $\theta_{\mathrm{MLP}}$, capable of approximating a variety of functions (details in Appendix B).

- An attention layer in the baseline transformer is defined as: $x \leftarrow \mathrm{softmax}(Q\,K^\top)\,V$, where $x$ is the output vector of activations and $Q, K, V$ are linear projections of the input. This is a special case of the kernel version of attention: $x \leftarrow \sum_{j=1}^n K(Q_i, K_j)\,V_j \,/\, \sum_{j=1}^n K(Q_i, K_j)$ where the similarity between $Q$ and $K$ is measured with a kernel function $K(Q, K)$. In the baseline transformer, $K_{\mathrm{smax}}(Q, K) = \exp(Q^\top K/\sqrt{d})$. In our model, we introduce a learnable non-linearity $\phi' : \mathbb{R} \to \mathbb{R}$ giving $K(Q, K) = \phi'(Q)^\top \phi'(K)$. We implement $\phi'$ as a linear spline $\phi'_{\theta_A}$ with keypoints $\theta_A$ that can be optimized.

**Two-stage setting.** Our experiments proceed in two stages. In stage I, we optimize the architecture for a chosen dataset $\mathbb{D}$ by training both the model's weights and its parametrized non-linearities $(\theta_A, \theta_{\mathrm{MLP}})$ on $\mathbb{D}$. In stage II, the non-linearities are frozen, and we retrain the model in a standard manner from scratch on any dataset $\mathbb{D}'$. The models obtained from stage II are thus fairly comparable with the baseline architecture.[2] When $\mathbb{D}' \neq \mathbb{D}$, i.e. a "mix-and-match" setting, stage II serves to evaluate whether the inductive biases optimized for $\mathbb{D}$ suit the learning of $\mathbb{D}'$.

**Optimizing architectures.** Our method may seem similar to prior work about learning activation functions (e.g. (Alexandridis et al., 2025)) but their goals are very different. These works seek to improve performance by continuously updating the activation during training. Whereas we seek to identify inductive biases that can remain hard-encoded in the architecture and further reused to train new models with other seeds and datasets (stage II). We make this possible with a **two-loss training**. During stage I, we hold out a fraction of the training data (e.g. 20%) that we use solely for optimizing the non-linearities, while we optimize the weights in a standard manner on the training set. This prevents a co-adaptation, that could cause the non-linearities to overfit particular weights or seed. This is particularly important for our experiments on algorithmic toy tasks, and even more so for improving length generalization[3] (Section 3.1). In this latter case, we hold out an out-of-distribution (OOD) split of data (see Section 3.1), such that the weights are optimized for one range of sequence lengths, and the architecture for a different wider range. This forces the architecture to capture an inductive bias for length generalization. In stage II, the non-linearities are frozen, and the model weights are trained in a standard manner on the whole training split of the target dataset.

A second innovation to prevent the co-adaptation of weights and non-linearities in stage I is **multi-model training**. We train $M$ models in parallel (e.g. $M = 4$) that use different seeds but share the non-linearities being optimized. The resulting optimized architecture is naturally more likely to generalize in stage II to other weights and datasets (see Appendix D). This also proves particularly helpful for algorithmic tasks because the variance across seeds of the baseline architecture is often high. We provide a complete description of our method as Algorithm 1 in the appendix.

---

[2]In stage II, $(\theta_A, \theta_{\mathrm{MLP}})$ are frozen and better viewed as pre-tuned hyperparameters than extra model capacity.
[3]The benefit of the two-loss training is smaller for language modeling because the models are heavily over-parametrized and never at risk of overfitting the training data.

**Rationale for splines.** We parametrize our non-linearities as linear splines because they offer the most unbiased, tractable parametrization for an $\mathbb{R} \to \mathbb{R}$ function. For example, a spline can represent the identity function as easily as a step function or a sine wave. Prior work on trainable activation functions uses e.g. small MLPs that enforce priors like smoothness or monotonicity (Apicella et al., 2021; Greydanus & Kobak, 2020). Such parametrizations would struggle to capture sharp transitions like those in Figure 6. We also favor *linear* splines over higher-order (e.g. cubic) ones because they behave similarly while being much cheaper, as evaluated by Teney et al. (2025, Appendix D).

## 3 EXPERIMENTS ON ALGORITHMIC REASONING TASKS

In this section, we apply the proposed method to a set of tasks commonly used to evaluate the algorithmic skills of transformers (Table 1). These tasks seem elementary but they are remarkably challenging for transformers, and often used to highlight their limitations. The tasks are all formulated as sequence completion: each example comprises an "input" part, followed by a separator, then an "output" part. The models are trained for next-token prediction on the output part of training sequences. Unless otherwise noted we use i.i.d. sets of training, validation, and test data.

**Experimental setup.** For each task $\mathbb{D}$, we first train the baseline architecture and tune its hyperparameters (width, depth, learning rate, batch size, etc.) for high accuracy and fast convergence on the validation set. We then run the proposed method (stage I, $M=8$) to optimize the architecture for $\mathbb{D}$. We then re-train a model from scratch with the optimized architecture (stage II), keeping the same hyperparameters (we get no further improvements by re-tuning them). In Section 3.2, we also re-train models on other tasks $\mathbb{D}'$ as a way to evaluate the compatibility between $\mathbb{D}$ and $\mathbb{D}'$, and the generality of the optimized architectures. All results are averages over 6 random seeds.

Table 1: Algorithmic tasks used in our experiments. They are sized such that they require similar model capacity, except for MANO (Allen-Zhu, 2025) which is intrinsically more complex.

| Task | Examples |
|---|---|
| **MEMORIZE**. Simple memorization of a mapping between a two-integer key and an integer value, with all integers in $[1,32]$. Each sequence consists of the key, a separator, and the value. This task has no test set: performance is simply the training accuracy (Zhong & Andreas, 2024). | `23 12 | 10`
`11 32 | 27`
`31 19 | 18` |
| **PARENTHESES**. Recognition of Dyck language. Each sequence contains parentheses followed by a separator and a marker indicating whether they are balanced or not. Sequences lengths are in $[1,20]$ in the training set, and $[21,40]$ in the validation and test sets (Zhong & Andreas, 2024). | `( ) ( | <unbalanced>`
`( ) ( ) ) | <balanced>`
`) ( ) ( ) | <unbalanced>` |
| **ADDMOD**. Modular addition mod $N$, with 95% of the $N^2$ examples used for training (Zhong & Andreas, 2024). We use $N$=97. | `12 3 | 15`
`96 2 | 1` |
| **HAYSTACK**. Needle-in-a-haystack recall. The model gets a sequence $[m_1, c_1 \dots m_k, c_k, m_u]$ of markers $m_k$ and values $c_k$. It must search for the first occurrence of $m_u$ and return its successor $c_u$ (Zhong & Andreas, 2024). We use $k \in [1,10]$ and $m_k, c_k \in [1,64]$. | `2 p 9 k 3 b 9 | k`
`8 a 2 b 8 | a`
`2 p 9 k 3 b 5 x 5 | x` |
| **ADD**. Decimal addition of 4-digit numbers with digit-wise tokens. (Zhong & Andreas, 2024). | `1 0 0 9 + 1 0 9 2 | 2 1 0 1` |
| **ADDREVERSED**. ADD with reversed numbers, known to be easier to learn (Lee et al., 2023) . | `9 0 0 1 + 2 9 0 1 | 1 0 1 2` |
| **COPY**. Repeating the input. Elementary but unsolved for length generalization (Cai et al., 2025). Tokens in $[1,8]$. Seq. lengths in $[2,10]$ for training, $[2,15]$ for validation, $[16,20]$ for testing. | `2 8 | 2 8`
`9 4 8 7 8 3 | 9 4 8 7 8 3` |
| **MANO**. Synthetic task proposed by Allen-Zhu (2025) to evaluate large pretrained models. Each sequence specifies nested arithmetic operations mod $N$ with number-level tokens. Our scaled-down version uses $N$=7 and a number of operations per sequences in $[1,3]$. | `(1*3)+4 | 0`
`(2-(6-1))*3 | 5`
`(3*(5-6))-1 | 3` |

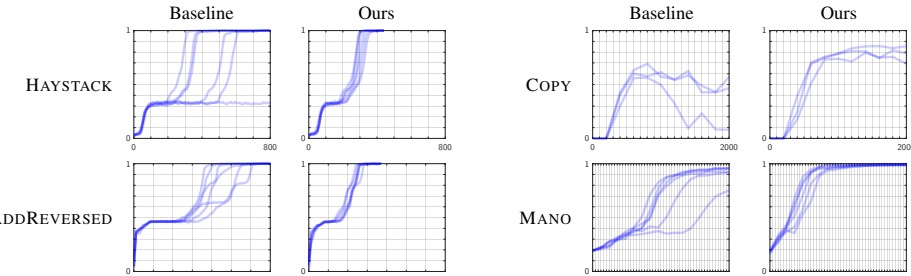

Figure 2: Training curves (test accuracy vs. training step, one curve per random seed) of models trained on algorithmic tasks with a baseline transformer or our optimized architectures. The latter converge much faster and show less variance across seeds. See Appendix C for other tasks.

### 3.1 IMPROVEMENTS ON INDIVIDUAL TASKS

**Faster convergence.** The most striking improvement with optimized architectures is the learning speed (Figure 2). For the ADD and MANO tasks for example, convergence occurs $2-3\times$ faster. The learning rate of the baseline was tuned to its maximum stable value for every task.

**Reduced variance.** On some tasks, baseline transformers show huge variance in accuracy and training speed across random seeds. This suggests tasks that are underspecified (Teney et al., 2021; 2022) and misaligned with the model's inductive biases (Zhou et al., 2024). In these cases, the optimized architectures eliminate the problem and make the training much more reliable (Figure 2).

**Better generalization.** For some tasks, baseline transformers do not reach perfect test accuracy though they perfectly fit the training data. This shows again a misalignment between the target function and the inductive biases. Optimized architectures solve this problem (see e.g. MANO, Figure 2).

**Improved length generalization.** An outstanding challenge for transformers is the generalization to sequences longer than seen during training. Even the COPY task is unsolved and a baseline transformer completely fails on unseen lengths (Figure 3). Among the plethora of existing partial solutions, the Alibi positional encodings (Press et al., 2021) bring non-trivial accuracy on slightly longer sequences. We use our method to optimize the Alibi architecture. We use the two-loss mechanism of Algorithm 1 to optimize the transformer weights on lengths 2–10 and the non-linearities on 2–15. This forces the optimized architecture to capture an inductive bias for length generalization. As a result, a model trained with the optimized architecture reaches higher accuracies on longer sequences. While this is not a complete solution to length generalization, it shows that inappropriate inductive biases in the base architecture are one of the obstacles to length generalization.

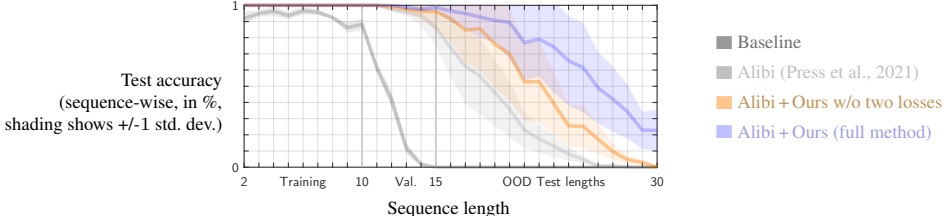

Figure 3: Length generalization on the COPY task. The baseline completely fails on unseen lengths ($\gg 10$). Alibi positional encodings (Press et al., 2021) help. Optimizing the Alibi architecture with our method further improves the accuracy and extends the benefits to longer sequences.

**Performance with smaller models.** We train models of different widths for each task. Results in Figure 4 show that the accuracy drops more sharply on some tasks with the baseline architecture than optimized ones. Intuitively, when the architecture is already aligned with the task, less capacity is needed in its weights. Equivalently, a fixed number of parameters offers more capacity.

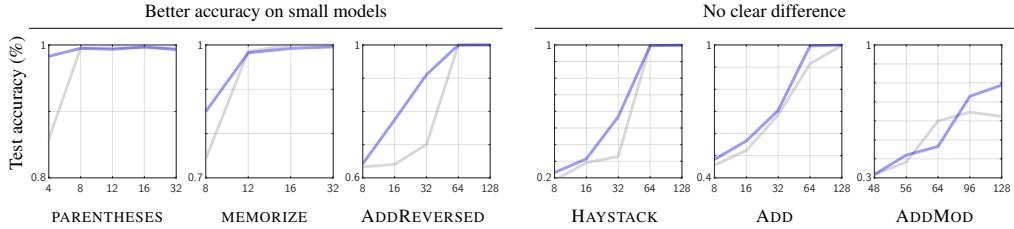

Figure 4: Test accuracy of models of different widths (X axis). On some tasks, optimized architectures (■) maintain higher accuracy than the baseline (■) when reducing the width of the model.

## 3.2 Compatibility of Optimized Architectures Across Algorithmic Tasks

We now train models on each task $\mathbb{D}$ using architectures optimized for any other task $\mathbb{D}'$ to evaluate the pairwise compatibility of their inductive biases. The results in Figure 5 show that the optimized architectures are very task-specific. Few of the benefits transfer across tasks, mostly across closely related tasks like ADD and ADDREVERSED. Many perform worse than a standard transformer. This shows that the specialization to our algorithmic tasks comes at the cost of universality. These tasks are very narrow however and it remains an open question whether the negative impact is inevitable. A future step to study this question could be a multi-task optimization in Algorithm 1.

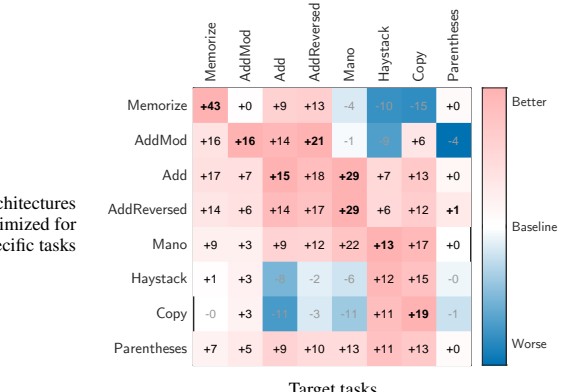

Figure 5: Compatibility of architectures across algorithmic tasks. We plot the absolute difference in test accuracy (%) with the baseline after a fixed number of steps (details in Appendix B). The best option per task (column) is usually on the diagonal, meaning that the optimized architectures are quite task-specific, while still yielding some positive transfer.

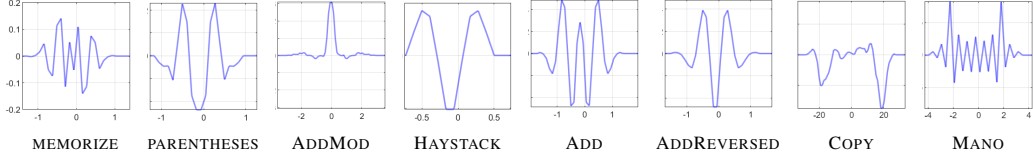

Figure 6: MLP non-linearities optimized for each algorithmic task.

> **Take-away.** On algorithmic tasks, optimized architectures can dramatically outperform standard transformers, but the benefits are quite task-specific. This means that these tasks require inductive biases very different from those of standard transformers.

## 4 Experiments on Language Modeling

We now apply the same experimental setup as Section 3 to language modeling. We use datasets for computer code (English, Java) and natural language of various complexity levels (Table 2). Our goal is to understand whether different types of data benefit from different inductive biases. Current practices for building LLMs show that data diversity is beneficial (Longpre et al., 2024) and that code is complementary to natural language (Aryabumi et al., 2024; Petty et al., 2024). But because all kinds of data are mixed during training, it is unknown whether they could each exploit or elicit different mechanisms in a model. We also consider versions of the datasets tokenized at the character or subword level (BPE; details in Appendix B). These choices are motivated by Mayilvahanan et al. (2025) who showed that LLM performance is mostly determined by data diversity and tokenization.

### 4.1 Improvements on Individual Datasets

**TINYSTORIES.** We compare in Figure 7 models trained with baseline or optimized architectures. The latter do slightly better. The improvement is small but consistent at different model sizes. Training curves (Figure 15) show that the improvement is larger early during training then diminishes. We

Table 2: Datasets used in our experiments for language modeling (see Appendix B for details).

| Dataset | Excerpt |
|---|---|
| **TINYSTORIES.** Children stories generated with GPT-3.5. It was designed to capture core aspects of natural language (syntax, coherence, compositionality) with a limited vocabulary. This allows smaller-scale experiments than web-scale open-domain corpora (Eldan & Li, 2023). | `Once upon a time, there was a clever little dog named Max.  Max loved to run (...)` |
| **SHAKESPEARE.** Plays and sonnets by William Shakespeare, often used in early research on language modeling. It includes recognizable patterns of grammar, rhythm, and vocabulary, as well as a unique structure because of the speaker labels and dialogue formatting (Karpathy, 2015). | `BENVOLIO: Good-morrow, cousin. ROMEO: Is the day so young? BENVOLIO: But (...)` |
| **ENWIK8.** First 100 M bytes of the English Wikipedia (Mahoney, 2006). We use the clean version from Yong (2025) with only text visible to human readers, without links and meta data. This data provides dense, real-world text with a mix of vocabulary, syntax, and formatting. | `anarchism originated as a term of abuse first used against early working (...)` |
| **CODESEARCHNET-JAVA & -PYTHON.** Dataset of computer code originally created to support research on code search and code–text understanding (Husel et al., 2019). We discard comments and descriptions in natural language following Lu et al. (2021) to focus exclusively on code. | `batch, limit = 100, self._next_limit() it = iter(it) (...)` |

Figure 7: **(Left)** Absolute improvements in token prediction accuracy (%) of the best optimized architectures on TINYSTORIES compared to our baseline transformer. The accuracy is consistently slightly better at different model sizes. **(Right)** Visualization of MLP non-linearities optimized from scratch (results on the left) or from a GeLU initialization (*GeLU + Ours* in Figure 8). Although they resemble generic wavelets, we show in Appendix D that fine details in these functions matter.

find it best to optimize non-linearities only in MLPs (i.e. replacing GeLUs; see Figure 8). Replacing softmaxes with learned components barely matches or underperforms the baseline, indicating a difficult optimization. We experimented with alternative parametrizations that exactly mimic a softmax at initialization. This solution would barely move away from this initialization (not reported in tables), suggesting that a softmax is close to a local optimum.

We visualize in Figure 7 (right) the optimized MLP non-linearities, which are remarkably similar to sine wavelets. We evaluate a non-exhaustive selection of activation functions and attention variants from the literature in Table 3. None of them works better than ours. The gated linear units (GLUs) are a popular design that adds multiplicative interactions to the MLPs. We show that we can also improve them by introducing our learned spline in GLUs in lieu of their internal Swish activations. This provides similar improvements as over standard MLPs, cf. GLU/*Swish* and GLU/*Ours* in Table 3. We also evaluate in Appendix D the importance of fine details in the learned non-linearities. We try to make them more periodic or symmetric, but they then always perform worse.

Table 3: Performance of models trained on TINYSTORIES with existing alternative attention and MLP designs (2 layers, width 256). None works better than ours. See Appendix D for references.

| Attention
MLP | smax
Linear | smax
GeLU | smax
Ours | smax
GLU/Swish | smax
GLU/Ours | smax
ReLU | smax
ReLU$^2$ | smax
TanH | smax
Sinc | smax
Gaussian | P1
GeLU | P3
GeLU | Adaptive
GeLU | NormSmax
GeLU |
|---|---|---|---|---|---|---|---|---|---|---|---|---|---|---|
| Tr. perplexity | 1.78 | 1.58 | **1.57** | 1.59 | 1.58 | 1.60 | 1.60 | 1.71 | 2.50 | 1.64 | 1.62 | 1.60 | 1.58 | 1.58 |
| Val. acc. (%) | 59.9 | 63.7 | **64.4** | 63.7 | 64.0 | 63.5 | 63.6 | 61.2 | 47.7 | 62.8 | 63.0 | 63.7 | 63.7 | 63.7 |

**SHAKESPEARE & ENWIK8.** These datasets differ from TinyStories in their richer vocabulary and sentence structure. SHAKESPEARE also follows a particular formatting presenting dialogues with speaker labels (see Table 2). The results in Figures 8 & 13 show that *some* optimized architectures slightly improve over the baseline. Optimizing non-linearities in the MLPs is again more useful than in the attention. However, differences with the baseline are small, which suggests that standard transformers are inherently well-suited to language modeling.

The improvement is slightly clearer on **character-level datasets** than on tokenized ones (marked -CHAR in Figure 8). We hypothesize that the target function to be learned by the transformer layers for character-level language modeling is more complex, because of the lesser capacity available

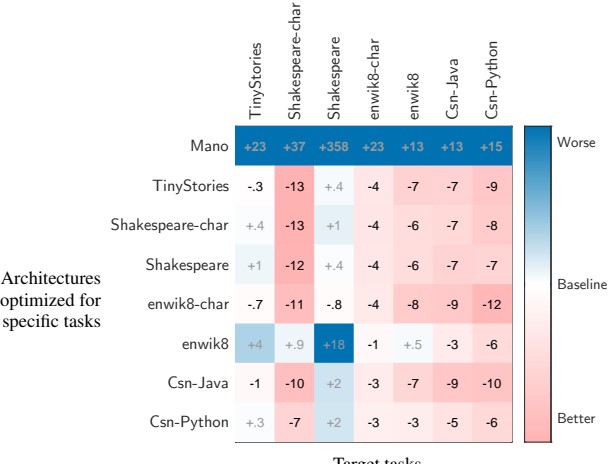

Figure 8: Perplexity on code and natural language (lower is better; numbers on bars correspond to the difference with the baseline architecture). Some optimized architectures perform slightly better than the baseline, often simply with optimized MLP non-linearities (■). Datasets of code (CSN-JAVA, CSN-PYTHON) also benefit relatively more than datasets of natural language.

in the model's token embeddings (embeddings can otherwise make up a significant fraction of the model parameters for tokenized datasets). This could be the reason why learned non-linearities are particularly helpful, since they can help learn and represent complex functions (Teney et al., 2025).

We also evaluate a version of our optimized **MLP non-linearities initialized as a GeLU** rather than a constant zero (*GeLU + Ours* in Figure 8). With this, the model starts stage I with a non-linearity known to perform well. And because the optimization is non-convex, the optimized solution remains in the local search space near GeLUs (see Figure 7, right). The models trained with these non-linearities perform in-between GeLUs and those optimized from scratch. This means that GeLUs are usually not an optimal solution, not even a local one. But note also that our best solutions are not guaranteed to be *globally* optimal and better ones may exist.

**CODESEARCHNET (CSN-JAVA, CSN-PYTHON).** The results in Figure 8 show that our optimized non-linearities in MLPs improve again over the baseline. The gains are larger for code than for natural language, relative to the gap between the baselines with linear and GeLU MLPs. These larger gains may reflect the larger importance of systematic structure and compositionality in code than natural language. The task of modeling code may thus resemble some of the algorithmic tasks of Section 3, which benefited greatly from optimized architectures. Therefore, the architectures best suited to natural language may not be simultaneously optimal for code.

## 4.2 COMPATIBILITY OF OPTIMIZED ARCHITECTURES ACROSS LANGUAGE DATASETS

Our final results examine the compatibility of the optimized architectures across language modeling datasets. We consider our seven datasets plus MANO, the most complex of our algorithmic tasks. We train models for every task $\mathbb{D}$ using architectures optimized for any other task $\mathbb{D}'$. The results in Figure 9 show that the variations across architectures are very small. This contrasts with the results on algorithmic tasks (Figure 5). These optimized architectures thus encode much less task-specific specialization. This suggests that the skills required across code and language modeling datasets are much more uniform. We discuss the implications of these results in Section 6.

Figure 9: Compatibility of architectures across code and language datasets (relative difference in perplexity with the baseline in %, lower is better). The differences are much less dramatic than with algorithmic tasks (Figure 5), indicating smaller benefit in dataset-specific specialization.

> **Take-away.** For code and natural language modeling, the optimized architectures improve much less than for algorithmic tasks. This means that standard transformers are intrinsically closer to a local optimum in the space of architecture for these tasks, than for algorithmic skills.

## 5 RELATED WORK

**Understanding inductive biases in NNs.** Much of the prior on understanding neural networks (NNs) has focused on their simplicity bias, i.e. their preference for representing functions of low Kolmogorov (Zhou et al., 2023) or spectral complexity (Bhattamishra et al., 2022). The simplicity bias depends primarily on the choice of activation function (Mingard et al., 2019; Teney et al., 2024), and its suitability was questioned (Domingos, 1999) by evaluating alternative activation functions in MLPs (Teney et al., 2024). We extend this inquiry to transformers and larger settings. In particular, we introduce a method to optimize non-linearities in both attention and MLP layers, and apply it to tasks relevant to the state of the art (code, natural language, algorithmic reasoning).

**Improving transformers.** Current LLMs all use very similar architectures, and Mayilvahanan et al. (2025) show that small design differences play little role in their performance. Prior work has however studied at length the impact of various components of transformers including their nonlinearities (Jha & Reagen, 2025; Newhouse et al., 2025). Proposed improvements include alternative attention mechanisms (Katharopoulos et al., 2020; Saratchandran et al., 2024b; Schlag et al., 2021; Tamayo-Rousseau et al., 2025; Veličković et al., 2024) and activation functions for MLPs (Hu et al., 2025; Teney et al., 2025) and transformers (Mirzadeh et al., 2023; So et al., 2021a). This motivates our work by suggesting that standard transformers are not a uniquely optimal choice of architecture.

**Architecture search.** Our method to optimize architectures is reminiscent of neural architecture search (NAS) (Goyal et al., 2019; Hong, 2025; Liu et al., 2018; Manessi & Rozza, 2018; Ramachandran et al., 2018; Zoph & Le, 2017). The goals and approach are different though. NAS uses RL or evolutionary algorithms to search through pre-defined design choices. We directly use gradient descent to optimize a relatively unrestricted parametrization of the non-linearities of transformers. Our goal is not to find better models (our designs are often computationally expensive). Instead, our method is a tool to understand the compatibility of the inductive biases required for various tasks.

## 6 DISCUSSION

We have presented a method to optimize a transformer architecture for specific datasets and used it to study the compatibility of inductive biases across tasks. We found that standard transformers are often suboptimal, but minor tweaks (replacing GeLUs and softmax operations) can substantially improve training speed, generalization, capacity, and stability across random seeds.

Our results show that different tasks benefit from different inductive biases, aligning with the no-free-lunch theorem (Wolpert, 1996). Yet, transformers seem uniquely suitable to a vast range of applications (Goldblum et al., 2023): our results can be seen as probing the limits of this hypothesis.

**Architecture vs. scale.** Prior work showed that the choice of architecture can become less important with scale (Bachmann et al., 2023; Tay et al., 2022). But this also means that the current need to build ever-larger models may be due to suboptimal inductive biases. In this work, we tweaked the non-linearities of transformers, but different inductive biases could be obtained by completely different means such as initializations (Shinnick et al., 2025) or optimizers (Pascanu et al., 2025).

**Do we need domain-specific models?** Our results show a higher compatibility across language and code than algorithmic tasks. The latter are often used to highlight limitations of transformers. If they really represent desirable capabilities in LLMs, perhaps new architectures are required to combine capabilities for both language and algorithmic skills. A future step could be to apply our method to optimize architectures for multiple tasks simultaneously. It would also be interesting to evaluate whether the improved architectures transfer to other domains such as vision and speech.

**Limitations.** First, our **search space of architectures** is limited. Complex forms of attention (Hashemi et al., 2025) or interactions like gated linear units (GLU, Shazeer 2020) cannot be represented in our formulation. Further gains are possible with a larger search space, but the optimization will also be more challenging. Second, the **scale of our experiments** is tiny relative to state-of-the-art LLMs. The effects of different architectures may vanish with more data, but improving data efficiency is a key objective of this line of work. So, effects at a small scale are intrinsically meaningful. Third, our architectures with optimized non-linearities can seem computationally costly. However, we describe **efficient implementations** in Appendix B and find that polynomial approximations of the learned splines make this a non-issue

Code is available at `https://github.com/idiap/lm-afs`.

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

## A  ADDITIONAL RELATED WORK

**Inductive biases in deep learning** are due to choices of architecture (Goyal & Bengio, 2022) and of the learning algorithm (optimizer, objective, regularizers Kukačka et al. 2017). We focus on the former. The simplicity bias has been studied from both aspects. Most explanations attribute it to loss functions (Pezeshki et al., 2021) and gradient descent (Arora et al., 2019; Hermann & Lampinen, 2020; Lyu et al., 2021; Tachet et al., 2018). But work on untrained networks shows that it can be explained with architectures alone (De Palma et al., 2019; Goldblum et al., 2023; Mingard et al., 2019; Teney et al., 2024; Valle-Perez et al., 2018). Teney et al. (2024) showed that the choice of activation function can modulate the simplicity bias. The **spectral bias** (Rahaman et al., 2019; Kalimeris et al., 2019) or frequency principle (Xu et al., 2019) is a related but different effect related to training dynamics: NNs approximate low-frequency components of the target function earlier during training with SGD.

**Simplicity bias in transformers.** The hypothesis of a simplicity bias in NNs has also been studied specifically in transformers. Hahn et al. (2021) shows that common models in NLP are biased to learn low-sensitivity functions. Bhattamishra et al. (2022) shows that transformers are more biased for simplicity than LSTMs. Dziri et al. (2023) examine large pretrained models and determine that that tend rely on shortcut learning on simple reasoning tasks. Zhou et al. (2023) focus on length generalization and show that transformers learn the shortest program in the RASP language that fits the training data –a specific form of the simplicity bias. Rende et al. (2024) study BERT-like models and find that they learn simple functions first during the course of training. Zhang et al. (2024) find that the scale of initialization can influence a transformer's learning of a generalizing or memorizing solution. Vasudeva et al. (2024) further study the bias of transformers for learning low-sensitivity functions using the NTK theory. Hahn & Rofin (2024) show that sensitive functions are hard to learn for transformers because they correspond to sharp solutions in their optimization landscape as a side-effect of the simplicity bias.

**Activation functions** are key for introducing non-linearities in NNs. Many options were considered early on, e.g. sine activations in the Fourier Neural Networks from 1988 (Gallant, 1988). ReLUs are often credited for enabling the rise of deep learning by avoiding vanishing gradients (Maas et al., 2013). However they are also essential in inducing the simplicity bias (Teney et al., 2024) which may be just as important. The research community has slowly converged towards smooth handcrafted variants of ReLUs such as GeLUs (Dubey et al., 2022; Hendrycks & Gimpel, 2016; Ramachandran et al., 2017). Some works proposed **learning activation functions** using extra parameters optimized alongside the weights of the network (Alexandridis et al., 2025; Apicella et al., 2019; 2021; Bingham et al., 2020; Chelly et al., 2024; Ducotterd et al., 2024; Jagtap et al., 2020; Scardapane et al., 2019; Sütfeld et al., 2020). See Jagtap & Karniadakis (2023) for a comprehensive review. The goal is to better fit the training data with an activation function that can evolve during training. In contrast, we use meta learning to find an activation function that induces better inductive biases, such that training with this *fixed* activation provides better generalization. This requires bi-level optimization, episodic training, and unbiased parametrization that allows us to learn activations very different from existing ones. **Kolmogorov-Arnold Networks** (Liu et al., 2024) parametrize the connections in a NN, which is equivalent to learning different activation functions across channels and layers. They use a parametrization as splines similar to ours. Their benefits in physics-related problems likely result from the alterations to the inductive biases studied in this paper. Our method differs from **neural architecture search** (White et al., 2023) in its ability to discover novel activation functions from scratch, rather than selecting from predefined candidates (Sütfeld et al., 2020) or from a narrow set of parametric functions (Alexandridis et al., 2025).

**Length generalization** refers to the ability of a model to generalize to sequences longer than seen during training, especially for algorithmic tasks (e.g. arithmetic operations on numbers with more digits). This remains a challenge despite extensive work on positional encodings, which only partially address the problem (Anil et al., 2022; Kazemnejad et al., 2023; Zhou et al., 2024). This paper shows that other aspects of the architecture can be important. We use the COPY task as proof of concept and show that different MLP activation functions can bring a significant improvement to the existing Alibi encodings (Press et al., 2021).

**Learnability and inductive biases.** The learnability of any given task is a fundamental question in machine learning. It is well know that inductive biases are indispensable for generalization to unseen data (Mitchell, 1980) and that no learning algorithm is universally useful, as per one of the no-free lunch theorems (Wolpert, 2002)). Meanwhile, neural networks have nevertheless proved widely successful. The broad applicability of transformers, in particular, suggests that their inductive bias has a broad relevance to real-world data (Goldblum et al., 2023). The **simplicity bias** is a broad and vague characterization of these properties. Various studies have established however that the simplicity bias is not universally beneficial (Domingos, 1999; Teney et al., 2025; Zeng et al., 2023) and even responsible for failure cases such as **shortcut learning** (Geirhos et al., 2020; Puli et al., 2023; Teney et al., 2021) or the amplification of biases and performance disparities (Bell & Sagun, 2023). Even the underlying principle supporting the simplicity bias, known as **Occam's razor**, has long been debated in the philosophical literature because it lacks a justification from first principles (Mingard et al., 2023, Appendix A). A prominent argument for simplicity is rooted in algorithmic information theory (Dingle et al., 2018) with results stating essentially that "*a bias in the distribution of target functions must be towards low complexity*". However, this only means that simplicity is a good prior on average, but not necessarily the best choice for any task or dataset.

Studies in linguistics and cognitive science have also examined the question of learnability. This includes studies on the influence of architectures and data on generalization during **language acquisition**, both for humans and machines (Futrell & Mahowald, 2023; Millière, 2024; Warstadt & Bowman, 2020). This explains how syntactic and structural biases arise and how they can be controlled (Mueller & Linzen, 2022; Papadimitriou & Jurafsky, 2022; Yang et al., 2024). Our paper complements this line of work since it helps clarify the impact of architectures on generalization. Our approach is quite different though. Our method allows searching through the space of architectures via the optimization of non-linearities. This matters because current popular designs (e.g. transformers) are contingent on external factors (see the *Hardware Lottery*, Hooker 2021).

**Connection with prior work.** This paper is a follow-up the study by Teney et al. (2025) that uses trainable non-linearities to study whether the *simplicity bias* of standard neural architectures is always desirable. It was however limited to MLPs and toy data, and relied on an expensive optimization method unsuitable to modern architectures. In comparison, our main innovations are:

- a formulation of trainable non-linearities that applies to transformers' MLPs and attention layers;
- a tractable optimization method replacing the expensive bi-level approach from prior work;
- the study of mainstream domains (language modeling, algorithmic reasoning);
- the study of cross-task compatibility, whereas prior work focuses on individual datasets;
- a demonstration of massive improvements on algorithmic tasks;

# B   IMPLEMENTATION DETAILS

**Proposed method.** We provide a formal description of our method in Algorithm 1.

---

**Algorithm 1** Proposed method (stage I) to optimize a transformer architecture for a specific task.

---

**Input**:

Training data $\mathbb{D} = \{s_i\}_{i=1}^n$ as token sequences $s \in \mathbf{S}$.

Baseline architecture $\mathcal{T}$ instantiable as next-token prediction model $T_{\boldsymbol{\theta}} : \mathbf{S} \to \mathbf{S}$ of weights $\boldsymbol{\theta}$.

$\mathcal{L}(\cdot, \cdot)$: Loss function.    $\alpha$: Fraction of held-out data.    $M$: Number of parallel models.

**Method**:

Define a new architecture $\hat{\mathcal{T}}_{\boldsymbol{\theta}_{\mathrm{A}} \boldsymbol{\theta}_{\mathrm{MLP}}}$ by replacing
  - softmaxes with $\Sigma_j K(\mathbf{Q}_i, \mathbf{K}_j) \mathbf{V}_j / \Sigma_j K(\mathbf{Q}_i, \mathbf{K}_j)$, where $K(\mathbf{Q}, \mathbf{K}) = \phi_{\boldsymbol{\theta}_{\mathrm{A}}}(\mathbf{Q})^\top \phi_{\boldsymbol{\theta}_{\mathrm{A}}}(\mathbf{K})$.
  - GeLUs with a linear spline $\phi_{\boldsymbol{\theta}_{\mathrm{MLP}}}$,

where architecture hyperparameters $\boldsymbol{\theta}_{\mathrm{A}}$ and $\boldsymbol{\theta}_{\mathrm{MLP}}$ specify the value of splines $\phi$ at their keypoints.

Instantiate $M$ untrained models of architecture $\hat{\mathcal{T}}$ as $\hat{T}_{\boldsymbol{\theta}_1}^1 \dots \hat{T}_{\boldsymbol{\theta}_M}^M$.

Split $\mathbb{D}$ into $\mathbb{D}_{\mathrm{arch}}$ and $\mathbb{D}_{\mathrm{wts}}$ of sizes $\alpha n$ and $(1-\alpha)n$.

> **while** not converged   *SGD training loop*
>
> Sample mini-batch $\mathbb{D}^0$ from $\mathbb{D}_{\mathrm{arch}}$  and $\mathbb{D}^1 \dots \mathbb{D}^M$ from $\mathbb{D}_{\mathrm{wts}}$
>
> Eval. loss of each individual model on its own data $\mathbb{D}^m$:  $L_{\mathrm{wts}}^m \leftarrow \Sigma_{s \in \mathbb{D}^m} \mathcal{L}(\hat{T}_{\boldsymbol{\theta}_m}^m(s), s)$
>
> Eval. combined loss of all models together on $\mathbb{D}^0$:    $L_{\mathrm{arch}} \leftarrow \Sigma_m \Sigma_{s \in \mathbb{D}^0} \mathcal{L}(\hat{T}_{\boldsymbol{\theta}_m}^m(s), s)$
>
> Update weights of each model: $\forall m, \boldsymbol{\theta}_m \leftarrow \mathrm{SGD}(\boldsymbol{\theta}_m, \nabla_{\boldsymbol{\theta}} L_{\mathrm{wts}}^m)$
>
> Update architecture: $(\boldsymbol{\theta}_{\mathrm{A}}, \boldsymbol{\theta}_{\mathrm{MLP}}) \leftarrow \mathrm{SGD}((\boldsymbol{\theta}_{\mathrm{A}}, \boldsymbol{\theta}_{\mathrm{MLP}}), \nabla_{(\boldsymbol{\theta}_{\mathrm{A}}, \boldsymbol{\theta}_{\mathrm{MLP}})} L_{\mathrm{arch}})$

$(\boldsymbol{\theta}_{\mathrm{A}}^\star, \boldsymbol{\theta}_{\mathrm{MLP}}^\star) \leftarrow (\boldsymbol{\theta}_{\mathrm{A}}, \boldsymbol{\theta}_{\mathrm{MLP}})$.

**Output**: optimized architecture $\hat{\mathcal{T}}_{\boldsymbol{\theta}_{\mathrm{A}}^\star \boldsymbol{\theta}_{\mathrm{MLP}}^\star}$

*Now $\hat{\mathcal{T}}$ can be used like any other architecture, treating $\boldsymbol{\theta}_{\mathrm{A}}^\star$ and $\boldsymbol{\theta}_{\mathrm{MLP}}^\star$ as fixed hyperparameters.*

---

**Baseline transformer architecture.**  Our baseline is a GPT-2-style architecture (Radford et al., 2019). It uses standard multi-head attention, GeLU activation functions in the MLPs, post-norm layers, learned absolute positional embeddings, and a width multiplier of $4$ in the MLP hidden layers. All weights are initialized from Gaussians of standard deviation $0.02$ truncated at 2 standard deviations.

**Parametrization of non-linearities as linear splines.**  We want a search space free of priors such as the smoothness and monotonicity enforced in similar work on the learning of activation functions (e.g. Apicella et al. (2019); Chelly et al. (2024)). We therefore choose to learn a non-linearity as a linear spline $\phi_{\boldsymbol{\theta}} : \mathbb{R} \to \mathbb{R}$ with control points defined by $\boldsymbol{\theta}$. We define $n_{\mathrm{c}}$ points spread regularly in an interval $[a, b]$, typically $n_{\mathrm{c}} = 122$ points in $[-20, +20]$ for a spacing of $1/3$ between points (see hyperparameters in Table 4). Then $\phi$ represents piecewise linear segments interpolating values specified in the learned parameters $\boldsymbol{\theta} := [\phi_{\boldsymbol{\theta}}(a), \dots \phi_{\boldsymbol{\theta}}(b)] \in \mathbb{R}^{n_{\mathrm{c}}}$. The function $\phi$ can represent simple and complex functions, including smooth curves, periodic functions, sharp transitions, etc.

**Datasets for algorithmic tasks.**  For most tasks, we generated data with code adapted from Zhong & Andreas (2024): `https://github.com/fjzzq2002/random_transformers`. While this prior work generates some of the data on-the-fly, we pre-generate all the data to ensure that the training/validation/test splits are strictly disjoint.

For MANO, we re-implemented the data generation based on the description by Allen-Zhu (2025). Compared to this prior work, we scaled down the task to allow using smaller models. We generated 1e5 training examples, with a number of operations in each sequence in $[1,3]$, a modulus of 7, and without tokens signaling the number of operations.

For all algorithmic tasks, we use a test set of 1e3 examples, strictly disjoint from the training set.

**Datasets for language modeling.** For datasets tokenized at the character level, every character or symbol in the data simply corresponds to one token. For the TINYSTORIES, SHAKESPEARE, and ENWIK8 datasets tokenized at the subword level, we use the byte-pair encoding (BPE, Gage (1994)) tokenizer from GPT-2 Radford et al. (2019). We consider it a consistent choice suitable to our different datasets since it was originally trained on very diverse data. For the CODESEARCHNET datasets, we use the tokenizer of the CodeGPT model (CodeGPT, 2024). For each dataset, we discard tokens with fewer than 200 training occurrences. This significantly reduces the vocabulary size and training costs. This should not undermine the results of our experiments: if anything, including more rare tokens could reveal larger differences across datasets.

**Metrics.** For the algorithmic tasks, we measure performance as the token-wise accuracy of the "output" part of the generated sequences (the same part of the sequences as used to compute the training loss). This allows a finer-grained evaluation of partial success than the sequence-wise accuracy.

For the COPY task, we use the sequence-wise accuracy because the token-wise accuracy can remain falsely high when a model fails at length generalization.

For the language modeling tasks, we measure performance using the training perplexity (exponential of cross-entropy loss) as well as token-wise accuracy on validation data as a more intuitive measure of performance. For the accuracy, we measure it on the latter half of the context window to ensure that we evaluate predictions with enough conditioning.

For the compatibility across algorithmic tasks (Figure 5), we plot the difference in test accuracy with the baseline after a fixed number of steps. We adapt the number of steps to each task to capture improvements in generalization and/or training speed depending on the task. This is because both the baseline and optimized architectures saturate at perfect accuracy for multiple tasks, hence the *final* accuracy alone is not informative.

- MEMORIZE: 150 steps.
- PARENTHESES: 300 steps.
- ADDMOD: 300 steps.
- HAYSTACK: 400 steps.
- ADD: 700 steps.
- ADDREVERSED: 350 steps.
- COPY: 2,000 steps.
- MANO: 3,000 steps.

**Hyperparameters.** We tuned the hyperparameters in Table 4 for a standard transformer on each task, to make sure that our optimized architectures are compared against strong baselines. For example, we use the Canon layers proposed by Allen-Zhu (2025) for many tasks (sequence-wise 1D convolutions) because they clearly improve the performance of the baseline.

Table 4: Hyperparameters used for each task.

| | MEMORIZE | PARENTHESES | ADDMOD | HAYSTACK | ADD | ADDREVERSED | COPY | MANO | Language datasets |
|---|---|---|---|---|---|---|---|---|---|
| Num. layers | | | | 2 | | | | | 4 |
| Num. att. heads | 2 | 2 | 2 | 2 | 2 | 2 | 8 | 4 | 4 |
| Width | 32 | 32 | 32 | 128 | 128 | 128 | 128 | 128 | 512 |
| Tied embeddings | | | | No | | | | | Yes |
| Canon layers | | | | No | | | | Yes | |
| Num. tr. steps | 500 | 500 | 1,000 | 1,000 | 1,000 | 1,000 | 2,000 | 5,000 | 3,000 |
| Peak LR | .005 | .001 | .02 | .001 | .001 | .001 | .004 | .001 | .001 |
| Batch size | | | | 512 | | | | | 64 |
| Optimizer | | | | Adam | | | | | |
| Adam $(\beta_1,\beta_2)$ | | | (0.9, 0.999) | | | | (0.92, 0.98) | | (0.9, 0.999) |
| LR schedule | | 5% linear warm-up, 50% cosine cool-down (not necessary for algorithmic tasks; used on all tasks for consistency) | | | | | | | |
| Weight decay | | | 0 (better on all tasks than using any weight decay) | | | | | | |
| Dropout rate | | | | 0 | | | | | |
| Parallel models $M$ | | | | 8 | | | | | 3 |
| Spline range $[a,b]$ | | | | $[-20, 20]$ | | | | | |
| Spline spacing $n_{\text{c}}$ | | | 1/9 | | | | | 1/3 | |

## C  ADDITIONAL RESULTS ON ALGORITHMIC TASKS

**Training curves.** Figure 10 shows that the optimized architectures (2nd and 3rd columns) always converge significantly faster than a baseline transformer (1st column) and show less variance across seeds. There is little difference between the 2nd and 3rd columns, which means that most of the benefits come from optimizing the non-linearity within the MLP layers rather than the attention.

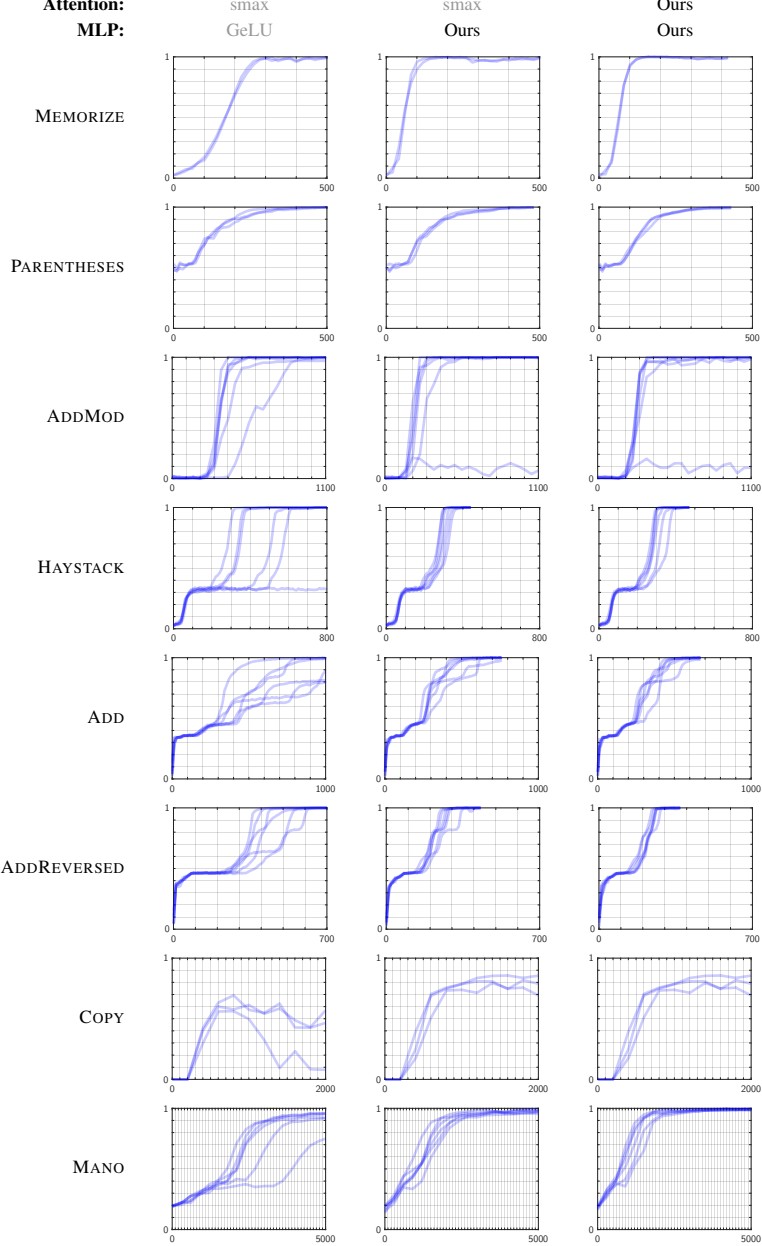

Figure 10: Training curves (test accuracy vs. training steps, one curve per seed) of models trained on algorithmic tasks with a baseline transformer (first column) or optimized architectures (second and third columns).

**Compatibility of architectures across algorithmic tasks.** We present below the full results following the format of Figure 5. We show the effect when optimizing the non-linearities in MLP or attention layers, or both. Optimizing the non-linearities in the attention proves to be really challenging, and the best results are usually obtained by optimizing only the MLPs.

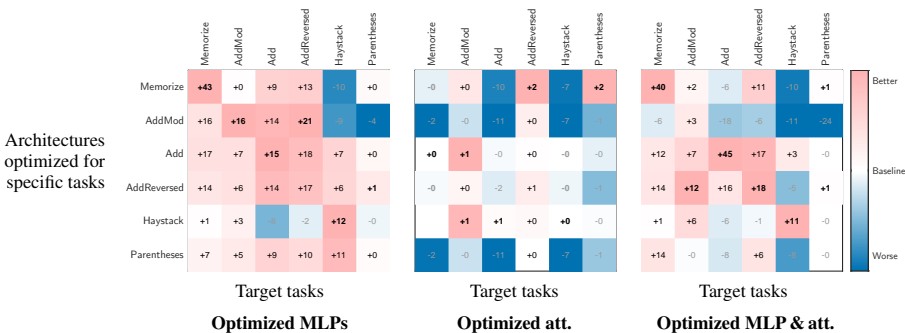

Figure 11: Compatibility of architectures across algorithmic tasks (difference in test accuracy with the baseline after a fixed number of steps).

# D   ADDITIONAL RESULTS ON LANGUAGE MODELING

**Manipulating optimized non-linearities.** In these experiments, we slightly modify the optimized MLP non-linearities to understand the importance of their fine details. Since they often look like sinusoidal wavelets, perhaps an even more regular version of them could perform better. We automate a "cleaning" process of the optimized non-linearities as follows. We take the optimized spline, reverse it along the X and/or Y axis (yielding three different versions), then align it with the original one by maximizing their cross-correlation. We then keep the average of the two. Among the three versions, we retain the one with the highest cross-correlation (i.e. similarity) with the original spline. The result is symmetric or anti-symmetric with fewer irregularities than the original one. We visualize this effect in Figure 12 on MLP non-linearities optimized for TINYSTORIES and various model sizes. We train models with these, but in almost every case, they perform worse than the original ones. This shows that fine details in the original optimized non-linearities matter.

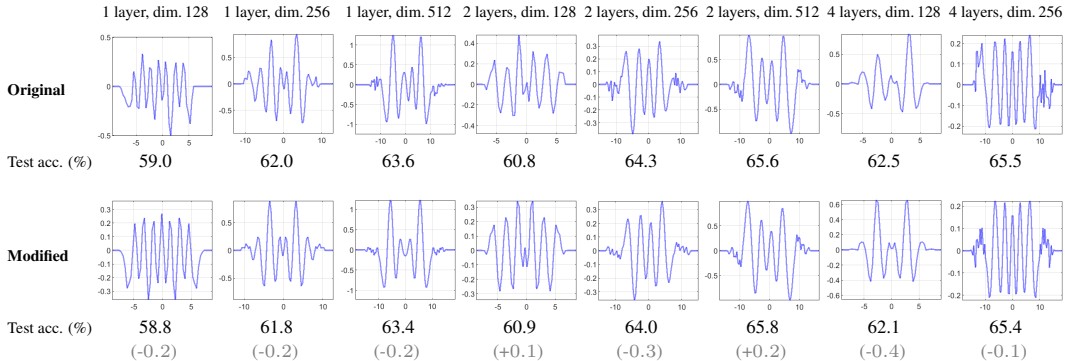

Figure 12: MLP non-linearities optimized for TINYSTORIES and versions modified to enforce symmetry. Almost all of these perform worse than the original ones, whose fine details therefore matter.

**Multi-model training.** We compare in Table 5 architectures for TINYSTORIES obtained with the proposed method and $M=1$ or $M=6$ models in parallel. The latter are slightly better, and the optimized non-linearities look slightly more regular.

Table 5: Models for TINYSTORIES with architectures optimized with $M$=1 or 6 parallel models.

| (Models with 2 layers, width 256) | **Attention** **MLP** | smax Linear | smax GeLU | smax Ours, $M$=1 | smax Ours, $M$=6 |
|---|---|---|---|---|---|
| | Tr. perplexity | 1.78 | 1.58 | 1.59 | **1.57** |
| | Val. acc. (%) | 59.9 | 63.7 | 63.8 | **64.3** |

| (Models with 4 layers, width 256) | **Attention** **MLP** | smax Linear | smax GeLU | smax Ours, $N$=1 | smax Ours, $N$=6 |
|---|---|---|---|---|---|
| | Tr. perplexity | 1.73 | 1.53 | 1.53 | **1.52** |
| | Val. acc. (%) | 60.8 | 65.1 | 65.3 | **65.4** |

**Existing methods.** Below are references for the attention and MLP designs evaluated in Table 3.

- **Adaptive softmax**: Veličković et al. (2024).
- **NormSoftmax**: Jiang et al. (2023).
- **Polynomial attention P1**: $(\mathbf{Q}^\top\mathbf{K})/\sqrt{\mathrm{seqLength}}$: Saratchandran et al. (2024b).
- **Polynomial attention P3**: $(\mathbf{Q}^\top\mathbf{K})^3/\sqrt{\mathrm{seqLength}}$: Saratchandran et al. (2024b).
- **GLU**: Shazeer (2020).
- **ReLU$^2$**: So et al. (2021b).
- **Sinc**: Saratchandran et al. (2024a).
- **Gaussian**: Saragadam et al. (2023).

**Full results on Shakespeare.** We present below results on the SHAKESPEARE dataset for various model sizes, in the same format as Figure 7. The best configuration is to optimize the MLP non-linearities while keeping the original softmax attention (second panels from the left).

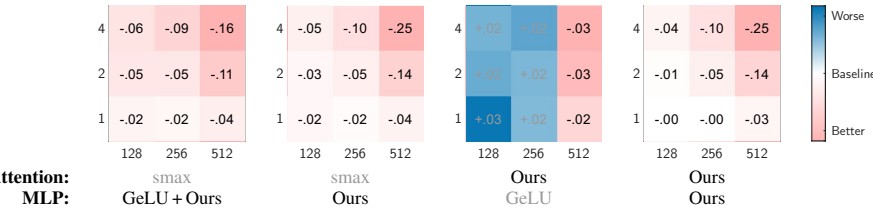

Figure 13: Absolute improvements in training perplexity on character-level SHAKESPEARE for models of different sizes (number of layers × width).

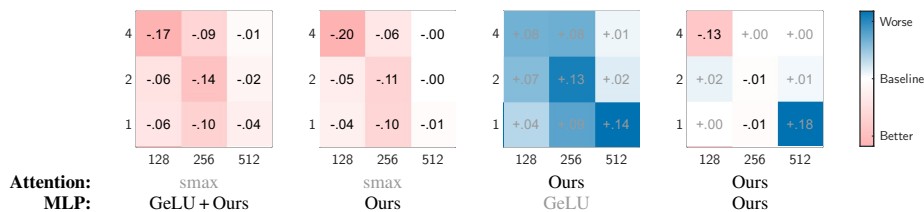

Figure 14: Same as Figure 13 with subword-level tokenization.

**Training curves on language datasets.** Figure 15 shows that the optimized architectures (■) show a larger improvement over a baseline transformer early during training, which then diminishes.

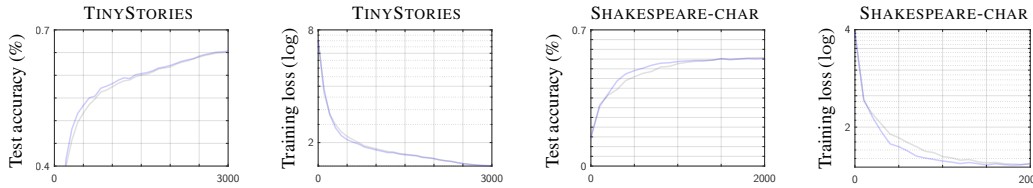

Figure 15: Training curves on language datasets with baseline (■) and optimized (■) architectures.

# E    RESULTS WITH LARGER LANGUAGE MODELS

On the suggestions of reviewers, we performed additional experiments to evaluate the improvements from the optimized non-linearities at various scales. We repeat experiments on language modeling as in Section 4 with the following differences.

- We use the **FINEWEB dataset** (Penedo et al., 2024), a popular high-quality dataset of cleaned and deduplicated English text from CommonCrawl.

- We implement our method on top of a very strong baseline, the **NanoGPT Speedrun** (Jordan et al., 2024). This is a competitive repository where contributors specifically push the implementation and data efficiency of the model on the FINEWEB dataset. We specifically build on top of record #16, which includes rotary embeddings, QK normalization, the Muon optimizer, sliding-window attention, mixed-precision training, etc. The code was designed for 8 H100 GPUs but we adapted it to enable experiments with a single Nvidia RTX 4090 laptop GPU. Our results are therefore not directly comparable with the official Speedrun competition.

- We first run stage I of our method to optimize the MLP non-linearities of a small model, since this stage is computationally more expensive (2 layers, width 256, 4 attention heads). We then re-use the optimized non-linearity to run stage II (i.e. standard training) with models of **various sizes from 2 to 12 layers**. This setup therefore evaluates how the optimized non-linearities transfer across models of different depths.

- We train similar models (with 2 to 12 layers) with a ReLU, which is the best baseline for this codebase. We always use a standard attention with a softmax since we found in Section 4 that it was difficult to improve upon.

**Results**. The results in Table 6 show that our optimized non-linearities perform similarly or better than the baselines. There is little improvement at the smallest scale (probably because the model is very weak overall) but we get a consistent improvement at all other scales, surpassing both the ReLU and GeLU baselines in most cases.

Regarding the computational cost of the optimized non-linearities, our implementation (Listing 1) is as fast or faster than a ReLU in very small models. In larger models however, they become much more expensive. We propose in Appendix F a polynomial approximation. Table 6 shows that this approximation performs about as well as the original spline and about as fast as a ReLU.

Table 6: Evaluation of models of various depths trained on FINEWEB (average over 3 seeds).

| | **Number of layers** | 2 | 4 | 8 | 10 | 12 |
|---|---|---|---|---|---|---|
| | Number of parameters (M) | 91 | 105 | 133 | 148 | 162 |
| Validation loss (FINEWEB) | Linear | 4.21 | 4.05 | 3.94 | 3.91 | 3.89 |
| | GeLU | **4.00** | 3.87 | **3.72** | 3.75 | 3.72 |
| | ReLU | 4.01 | 3.87 | 3.78 | 3.75 | 3.72 |
| | Ours: linear spline | **4.00** | **3.82** | **3.72** | **3.70** | 3.68 |
| | Ours: polynomial approx. ($n=18$) | 4.02 | **3.82** | 3.73 | 3.71 | **3.69** |

| | **Number of layers** | 2 | 4 | 8 | 10 | 12 |
|---|---|---|---|---|---|---|
| Training time (sec) | Linear | **1,440** | **1,920** | **2,940** | **3,540** | **19,680** |
| | GeLU | **1,440** | **1,920** | 3,090 | 20,580 | 34,020 |
| | ReLU | 1,500 | 1,980 | 3,120 | 13,080 | 28,020 |
| | Ours: linear spline | 1,500 | 2,070 | 8,520 | 26,700 | 81,720 |
| | Ours: polynomial approx. ($n=18$) | **1,440** | 2,040 | 3,180 | 14,070 | 29,100 |

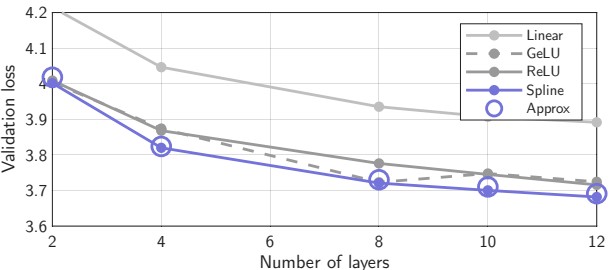

## F  EFFICIENT IMPLEMENTATION OF SPLINES

**Exact implementation.** Our non-linearities are parametrized as linear splines. We first provide an exact efficient implementation (Listing 1) that we find to be as fast as standard activations such as GeLUs for small models. However, depending on the architecture and GPU used, this function can quickly get bandwidth-constrained and become significantly slower. Therefore we propose a faster approximation with polynomials to be used when the spline has already been optimized and is used as a frozen non-linearity (i.e. for standard training, as in stage II of our experiments).

```python
# Evaluate at points x a function defined as the linear interpolation of knots
# of coordinates 'knotPos' and values 'knotVals'.
@torch.compile(dynamic=False)
def eval_spline(x, knotPos, knotVals):  # x: bfloat16, knotPos/knotVals: float32
    idx = torch.bucketize(x, knotPos) - 1  # Find the interval each x falls into
    idx = idx.clamp(0, len(knotPos) - 2)
    stepSize = knotPos[1] - knotPos[0]  # Interval between adjacent knots
    x0 = knotPos[0] + idx * stepSize  # Closest knot <= x
    frac = (x - x0) / stepSize  # Fractional position of x within the interval
    frac = frac.clamp(0.0, 1.0)  # Ensure constant extrapolation beyond the knots
    y0 = knotVals[idx]       # Value of preceding knot (<= x)
    y1 = knotVals[idx + 1]  # Value of following knot (> x)
    out = y0 + frac * (y1 - y0)  # Linear interpolation
    return out.to(x.dtype)  # For mixed precision; x can be bfloat16, NOT knotPos
```

Listing 1: Exact evaluation of a linear spline, used for stages I and II of most of our experiments.

**Approximation with polynomials.** The splines learned in our experiments with language models are quite smooth (unlike with algorithmic tasks in Section 3). It is therefore reasonable to approximate them with polynomials, which are much simpler and faster to evaluate. Concretely, given a linear spline optimized in stage I of our method, we determine an approximation through a least-squares fit of a polynomial of chosen degree $n$ on the spline values at its knots, on its support that has non-zero values. We choose a high degree ($n = 18$ typically) to ensure high fidelity with the original spline and to avoid ringing artifacts near the support boundaries. Beyond the boundaries, the polynomial is clamped to 0. For efficiency, we evaluate the polynomial with Horner's method, and implement it in a compiled function using `TorchsSscript` (see Listing 2).

```python
@torch.jit.script
def eval_polynomial(x: torch.Tensor) -> torch.Tensor:
    x = x.clamp(-79.52, 71.65) # Clamp for constant extrapolation
    x = x / 79.52 # Normalize to get values within [-1,1] for numerical stability
    return ((((((((((((((((((29327.20)*x + 18324.92)*x - 41591.43)*x - 12376.90)*x -
    14822.88)*x - 29015.27)*x + 33452.63)*x + 10354.57)*x + 21105.54)*x + 45592.25)*x -
    47565.33)*x - 47925.56)*x + 26296.37)*x + 18216.14)*x - 6145.61)*x - 2660.53)*x +
    522.13)*x + 66.86)*x - 0.63 # Evaluate polynomial with Horner's method
```

Listing 2: Example of a polynomial approximation, for the best spline from Table 7. It uses Horner's method with hard-coded coefficients and is compiled with `TorchScript` for efficiency. Note that this function was learned for the *NanoGPT speedrun* codebase which has unusually large internal activations, and is unlikely to directly work well with other architectures.

**Importance of *high degree* polynomials.** We tried reducing the maximum degree of the polynomials. This creates smoother functions that look appealing, but they perform systematically worse than high-degree polynomials or than the original spline. This shows the importance of fine details in the optimized splines. We also tried to suppress noise and artifacts near the support boundaries, by analytically enforcing null derivatives (up to 4th derivatives) of the polynomial at the boundaries. The functions are again visually appealing but they do not necessarily work better when training models with them. The data-driven optimization is clearly superior to our hand-crafted tweaks. One possible improvement that we have not implemented is an approximation with Chebyshev polynomials. These are known to provide better approximations of functions with finite supports, with less artifacts and better numerical stability.

**Do we need splines at all?** We tried to do away with splines entirely and directly optimize coefficients of a polynomial in stage I of our method. This completely fails however. Even though splines and polynomials can represent similar sets of functions, the different parametrization apply different inductive biases on the learned non-linearities. As discussed in Section 2, splines are particularly effective because they correspond to the most uniform prior on the space of functions.

**Evaluation of polynomial approximations.** We train small language models on FINEWEB with a different non-linearity for the MLP layers. We keep all hyperparameters identical and similar to Section E. Here, we use 6 layers, a width of 256, 4 attention heads, ∼20M parameters, and ∼80M training tokens. The results in Table 7 show that our spline performs best and slightly better than a ReLU. As expected, **the polynomial approximations are increasingly effective as we increase the degree**. The approximation then becomes very close to the exact spline. Low-degree polynomials yield smoother functions that are visually appealing but do not work as well. This shows that the parametrization as a spline is important to capture subtle important details.

Table 7: Models trained on FINEWEB with various MLP non-linearities. Our optimized spline works best. Approximations with high-degree polynomials are effective as they faithfully approximate the spline.

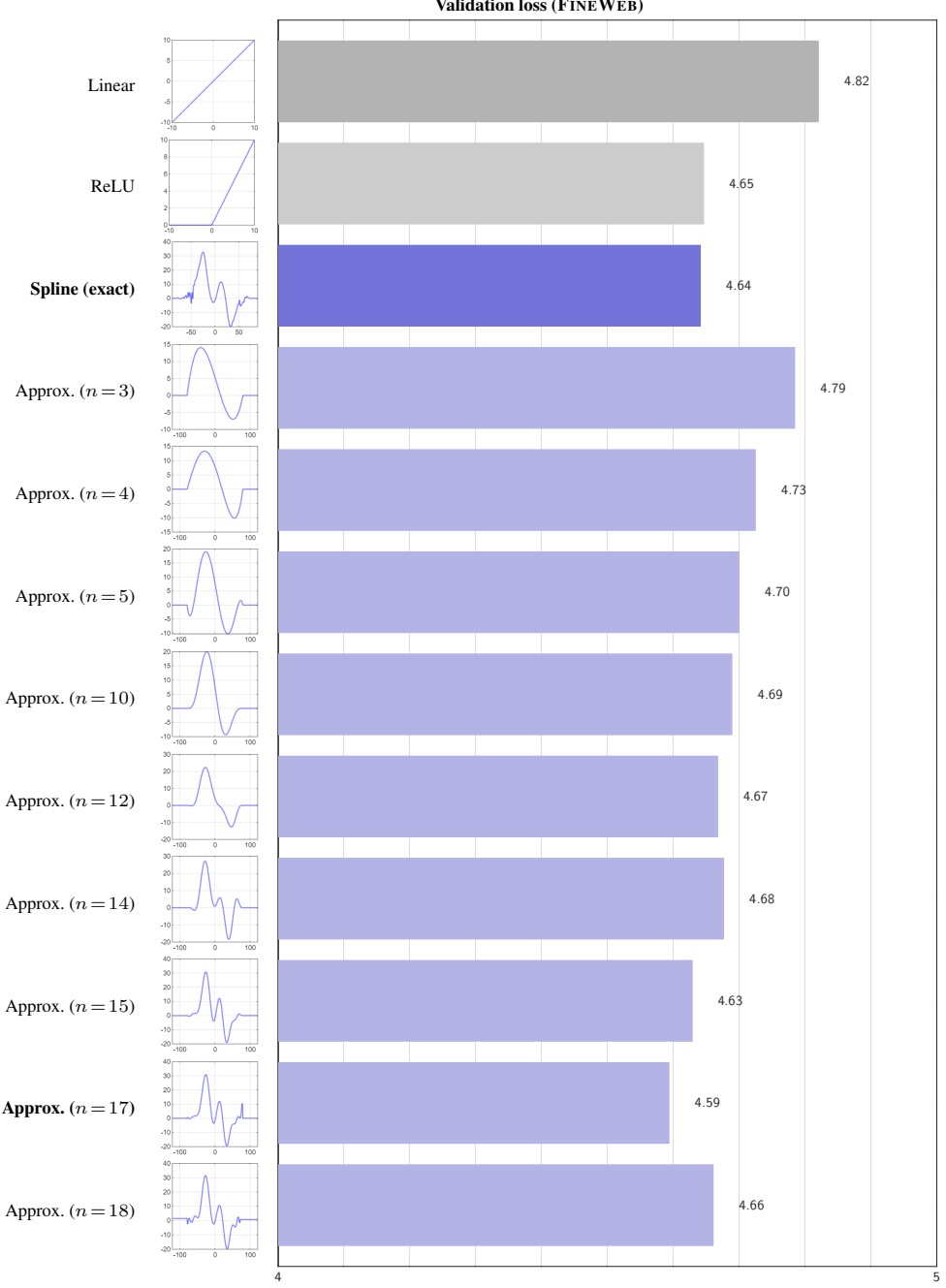

