# OpenReview forum: "Can Transformers Really Do It All? On the Compatibility of Inductive Biases Across Tasks"
_ICLR.cc/2026/Conference — ICLR 2026 Poster_

### Official Review · Reviewer_zxSg · 2025-10-27

**Soundness:** 2
**Presentation:** 3
**Contribution:** 1
**Rating:** 4
**Confidence:** 3

**Summary:**

The work investigates the question of whether transformers have the optimal architecture for all tasks. To investigate this, the authors replace the static nonlinearities of softmax in the attention block and the GeLU in the MLP block with trainable activations, and show that a 2 stage approach where we first train on a dataset to get the optimal shape of activation (the optimal architecture), and then in the second stage we fix the activation shape and then train on the dataset again from scratch, performs better than simply training a standard transformer architecture with standard activations. The benefits are in the form of convergence speed, stability across training seeds, and the final performance on held out test sets, as measured on algorithmic tasks as well as language modeling tasks. It also finds that usually the learnt activations that work best for one dataset usually are not the best choice for another datasets, suggesting dataset-specific variations in optimality.

**Strengths:**

* Good coverage of datasets for evaluation of the proposed hypothesis including both language modeling tasks and algorithmic tasks.
* The idea of using learnt splines for representing activation functions instead of fixed activations is creative.
* The presentation of the paper is good.

**Weaknesses:**

* Regarding the primary research question answered in the paper “Are transformers a unique and optimal solution endowed with generic inductive biases?” it is well known that one single architecture cannot be the best at learning all datasets and tasks, and there have been works in the past that have demostrated it too with respect to transformers (e.g. “Are Transformers Effective for Time Series Forecasting?” Zeng et al. AAAI 2023)
* The value of transformers’ unified architecture is that they can be pretrained to perform well across multiple tasks and domains. In this view the utility of having task specific architectures is diminished.
* The proposed approach uses 2-stage training where the first stage is used to learn activation function shape via parameterized splines and then training is restarted by training it from scratch but using the fixed learnt spline activation. This leads to faster convergence as shown in the paper. However, I am not sure if there is a fair comparison here because the first stage of activation learning will also be needed to be run on every dataset on which you want to train, and so the steps needed for the activation learning phase also need to be accounted for.

**Questions:**

* In section 3.1 and figure 1, the claim is that the model converges faster. Is the convergence steps measured by counting the training steps in stage2? This is relevant because this would mean that there is some information from stage1 training (the learnt spline parameters) that are re-used in stage2 and can thus lead to faster convergence.
* If one were to run the stage1 twice, do you get similar splines? I am wondering how much can we pinpoint to the learnt architecture to be optimal, given that there can be other splines that are just as good.

---

> ### Author Response · Authors · 2025-11-13
> **Response from the Authors**
>
> Thanks for the thoughtful review. We are adding the clarifications below to the manuscript. We also propose to clarify that **this paper is about a fundamental scientific question in ML** (*Are inductive biases compatible across tasks?*) rather than a method ready to train better LLMs.
>
> ----
>
> **W1: It is well known that one single architecture cannot be the best (...) Works in the past have demonstrated it with respect to transformers**
>
> We are indeed addressing a fundamental question in ML (compatibility of inductive biases across datasets) so there is a ton of related work, including Zeng et al. for time series forecasting. We propose to make this clearer in the *Related Work* section. Our contribution clearly brings new knowledge to the topic with a novel approach and methods, domains, and models studied.
>
> ----
>
> **W2: The value of transformers’ unified architecture is that they can be pretrained to perform well across multiple tasks and domains. In this view the utility of having task specific architectures is diminished.**
>
> This is indeed the status quo (for a variety of reasons) which does not mean this will always be desirable. The point of studying the compatibility of inductive biases across tasks is precisely to identify how much is left on the table by prioritizing a unified architecture.
>
> ----
>
> **W3: 2-stage training (...) not sure this is a fair comparison (...) the first stage will be needed on every dataset**
>
> The 2 stages are specifically designed as a fair comparison for the scientific questions being investigated, i.e. comparing inductive biases hard-coded in the architecture [1]. This setup is certainly not a prescription for training new models! Instead, this setup is designed to study the compatibility of inductive biases across datasets. Intuitively, if we can determine that a specific choice is effective across a family of datasets (e.g. language modeling), we'd have a new standard option for this domain (that does not require stage I).
>
> [1] Neural Redshift: Random Networks are not Random Functions, Teney et al. 2024
>
> ----
>
> **Q1: (...) information from stage1 (learned spline) re-used in stage2**
>
> Yes, this is exactly what the paper is about. Importantly, the evaluation in stage II uses data disjoint from stage I, so it's not about memorizing specific examples, but about identifying general inductive biases relevant to the task (the activation functions are parametrized with ~100 scalars so there is no capacity for memorizing examples).
>
> ----
>
> **Q2: stability of learned splines across runs**
>
> We observe little variance, especially because the proposed multi-model training is essentially averaging/ensembling across multiple seeds. Reassuringly, we do see variance like different runs learning the same spline flipped upside-down, left-to-right, or slightly offset on either axis. They are functionally equivalent and due to well-known weight space symmetries.
>
> ----
>
> The weaknesses above seem more like remarks than deficiencies of the paper. Hence we would love to hear about any outstanding issues relevant to the rating given to the paper. Thanks!

---

### Official Review · Reviewer_A5zP · 2025-10-31

**Soundness:** 3
**Presentation:** 3
**Contribution:** 4
**Rating:** 6
**Confidence:** 4

**Summary:**

This work intends to answer the question: is the standard transformer architecture optimal for all tasks? This question is important because the transformer architecture is often ported, without modification, to a wide variety of tasks. This work addresses this question by presenting a novel method for tuning the inductive biases in the transformer architecture to a particular dataset, and then evaluating performance of these optimized architectures on other datasets. This work demonstrates that the optimal inductive biases for algorithmic tasks are quite heterogeneous, often resulting in only mild improvements over the baseline architecture when used for other algorithmic tasks. However, the standard architecture is much better suited for language modeling tasks, with only mild improvements conferred to small models by optimizing the architecture.

**Strengths:**

This work addresses an interesting and important outstanding question.

This work uses a variety of tasks, both synthetic and naturalistic, to demonstrate that the standard transformer architecture is not a globally optimal architecture (or even a locally optimal architecture; see GeLU initialization results).

The results regarding the effect of scale are interesting and situate these results with the field — they acknowledge that scale might render inductive biases less relevant, but also demonstrate that proper inductive biases might enable smaller models. Though, see weakness section for concerns/questions.

**Weaknesses:**

The results for the optimized architectures on the language datasets do not seem significantly different than the baseline for the subword tokenized results. These same results could be used to argue strongly for the optimality of the baseline transformer architecture for language modeling. Only by using the (fairly nonstandard) metric of token accuracy do differences present themselves. This is not a problem, and is indeed interesting! It is worth making this point more explicitly in the text, rather than emphasizing small differences between the optimized and baseline architecture.

The results in Fig 7 seem to contradict the point made regarding character-level transformers in the text. In text, it is argued that smaller models (referring to char-level transformers) are better served by optimizing nonlinearities. However, this is inconsistent with the results on the tinystories dataset, where larger models consistently benefit more from optimizing the nonlinearities. How do you interpret this apparent contradiction?

In footnote 2 on page 3, the authors mention that the nonlinearities are better viewed as pre-tuned hyperparameters rather than extra model capacity, as they are frozen for stage 2 training. It would be very helpful to make this point empirically. For example, it would be great to have a baseline where the model is trained on all data in Stage 1, and performance is recorded from this model. If performance is higher in this setting than in the reported results, then one can feel more confident that the two stage training process is not merely adding extra capacity to the model.

It would be very interesting to compare modalities. In particular, characterizing how the optimal architecture for encoder-only models in language and vision differs would be very impactful.

**Questions:**

For Figure 4, is the caption mixing up the colors?

---

> ### Author Response · Authors · 2025-11-14
> **Response from the Authors**
>
> Thanks for the detailed review, the comments are extremely helpful for improving the manuscript.
>
> ----
>
> **W1: (...) Results for optimized architectures on language datasets not significantly different than the baseline for subword tokenized results.**
>
> Very true. We now realize that the text in §4.1 might be slightly misleading. We propose to better align it with the discussion in §4.2 (*Compatibility across language datasets*) and rather highlight the support for the strong adequacy of baseline transformers for language modeling.
>
> ----
>
> **W2: Apparent contradiction with character-level transformers.**
>
> Multiple competing effects are likely at play: character-level modeling requires not only larger models (as noted) but also a different, possibly more complex decision function (because of the lesser representational capacity in the set of embeddings). The need to model a function that is less aligned with the architecture's simplicity bias might be a reason for the larger observed benefit, instead or in addition to the smaller scale alone.
>
> This should clearly have been discussed in the text: we propose to tone down the current hypotheses and incorporate the above caveats.
>
> ----
>
> **W3: (...) it would be great to have a baseline where the model is trained on all data in Stage 1, and performance is recorded from this model. If performance is higher in this setting than in the reported results, then one can feel more confident that the two stage training process is not merely adding extra capacity to the model.**
>
> Indeed this is exactly what we observe. I.e. running phase I and II under identical conditions (same data, same number of steps), phase I usually reaches a (very) slightly better performance as the weights and activation functions (AF) can co-evolve and vary throughout training. We don't dwell on performance in this setting because:
> - this is much slower than standard training (backprop through the spline is expensive),
> - hence it is not a fair comparison to standard (fixed) activation functions,
> - there is a separate branch of the literature focusing on such dynamic/learnable activation functions [1-5],
> - and most importantly it does not allow drawing conclusions about the (AF-induced [6]) inductive biases that are our object of inquiry.
>
> [1] *Stochastic activations*, Lomeli et al. 2025
>
> [2] *Optimizing Neural Network Effectiveness via Non-Monotonicity Refinement*, Biswas et al. 2025
>
> [3] *Trainable Highly-expressive Activation Functions*, Chelly et al. 2024
>
> [4] *Adaptive Parametric Activation*, Alexandridis et al. 2024
>
> [5] *Adaptive Rational Activations to Boost Deep Reinforcement Learning*, Delfosse et al. 2024
>
> [6] *Neural Redshift: Random Networks are not Random Functions*, Teney et al. 2024
>
> ----
>
> **W4: It would be very interesting to compare modalities.**
>
> Absolutely! This project initially planned to compare inductive biases in ViTs versus LLMs. We had so many interesting results for this paper that there was no room to cover ViTs. But we are working on it now!
>
> ----
>
> **Q1: Figure 4, mixed color**. Fixed.
>
> ----
>
> Let us know if you see any outstanding issues with this submission. Thanks!

---

### Official Review · Reviewer_2nXb · 2025-11-03

**Soundness:** 3
**Presentation:** 3
**Contribution:** 4
**Rating:** 8
**Confidence:** 4

**Summary:**

This paper uses learnable non-linearities of linear splines in transformers, and optimizes the transformers for a variety of tasks, both small algorithmic tasks and language/code modelling. They find that optimized models converge faster and more consistently on the datasets that they are trained on, but that transfer is variable, meaning that there is no global configuration that is best for all types of data (and that transformers are not the local optimum for any one modelling task)

**Strengths:**

I am not very familiar with the complete literature around supervised architecture search, but with my knowledge I would say that this is an original and interesting method.

Both the framing (around finding inductive biases and how they compare between tasks), and the results, are interesting contributions, and I feel that I have learned something from this paper.

The analyses are thorough and well-done — for example I liked the extra analysis of Appendix D looking in to whether cleaner wavelets are better.

**Weaknesses:**

I think this is a strong and interesting paper, and any weaknesses are relatively minor:

W1: I would have also liked to see a bit more of an analysis of what the fact that we are restricting the search space to linear splines might mean. I understand that it is unreasonable to expect a paper to have a larger search space, so I think it’s perfectly reasonable to restrict to linear splines. However, though the authors do state it as a limitation, they do not provide much evidence or discussion about what the effects of this limitation might be, and what this can tell us about training LMs. Are there any pilot experiments you might do on one or two settings with a different function search space, to demonstrate the different effects it might have?

W2: I think that one of the strongest aspects of this paper  are the ideas around inductive bias and learnability, but the discussion is not as nuanced as it could be. I think it would be useful to mention some of the more cognitive-science-leaning studies on learnability and inductive bias such as (in no particular order, and no need to cite all I just want to give a sense of what I mean)

Futrell, Richard, and Kyle Mahowald. "How linguistics learned to stop worrying and love the language models.”

Mueller, Aaron, and Tal Linzen. "How to plant trees in language models: Data and architectural effects on the emergence of syntactic inductive biases.”

Papadimitriou, Isabel, and Dan Jurafsky. "Injecting structural hints: Using language models to study inductive biases in language learning.”

Hooker, Sara. "The hardware lottery.”

Warstadt, Alex, and Samuel R. Bowman. "What artificial neural networks can tell us about human language acquisition."

Millière, Raphaël. "Language models as models of language.”

Kallini, Julie, et al. "Mission: Impossible language models.”

Yang, Xiulin, et al. "Anything Goes? A Crosslinguistic Study of (Im) possible Language Learning in LMs.”

**Questions:**

Is there any analysis, or hypothesis to be tested, that you think comes out of seeing how the optimized splines look? It’s quite interesting that the optimized splines  don’t look much like the nonlinearities that we usually use, and I think that might be a fertile avenue for further analysis that can come out of this paper. Does it have to do with the initialization, or is there something about those structures that is better? Is it misguided of us as a field that we tend to use more monotonic nonlinearities? I really appreciate the analysis in Appendix D

---

> ### Author Response · Authors · 2025-11-15
> **Response from the Authors**
>
> Thank you for such a constructive and encouraging review.
>
> ----
>
> **W1: Other search space than splines**
>
> We just realized that the paper is missing the rationale for splines, which we are adding to Section 2 (also covered in [1]).
>
> In short, a linear spline (with many knots) is the most unbiased tractable parametrization for a 1-D real → real function (maximum-entropy prior). All alternatives impose stronger structural assumptions. For example, splines can represent the identity function just as well as a step function, or a Gaussian, or a sawtooth. An alternative parametrization like a small ReLU MLP [2] inherits a smoothness bias and would struggle to capture sharp transitions (like the splines do for algorithmic tasks, Fig. 6). Other parametrizations [2,3,4] that enforce e.g. smoothness or monotonicity are quite arbitrary and would defeat our aim of discovering optimized choices from data.
>
> Regarding the choice of *linear* splines, our early experiments and [1, Appendix D] show that higher-order splines (e.g. cubic) behave nearly identically while being more expensive. And 0-order splines (i.e. nearest-neighbour interpolation [5]) proved useless because they do not have proper (non-zero) gradients.
>
> [1] *Do We Always Need the Simplicity Bias? Looking for Optimal Inductive Biases in the Wild*, Teney et al., 2025
>
> [2] *Scaling Down Deep Learning with MNIST-1D*, Greydanus and Kobak, 2024
>
> [3] *Adaptive parametric activation*, Alexandridis et al., 2024
>
> [4] *A survey on modern trainable activation functions*, Apicella et al., 2021
>
> [5] *Kafnets: Kernel-based nonparametric activation functions for neural networks*, Scardapane et al., 2019.
>
> [6] *Trainable Highly-expressive Activation Functions*, Chelly et al., 2024
>
> ----
>
> **W2: Cognitive-science-leaning studies on learnability and inductive bias**
>
> Thanks for the suggestion. We're very familiar with this line of work, and feared it might have been a stretch to our "nuts-and-bolts" approach. We are gladly adding a section to the related work.
>
> ----
>
> **Q1: Analysis/hypothesis about the optimized splines**
>
> Some partial existing explanations:
> - theoretical work (in lower dimensions with implicit neural representations) showing that (non-monotonic) wavelets can be optimal in some sense thanks to their localization in both spatial and frequency domains [7,8];
> - known links between the non-linearities and Fourier-based measures of complexity of trained models [1] and of random samples in parameter space [9] which helps if aligned with the properties of the target function [10].
>
> As noted by the reviewer, Appendix D shows that fine details in non-linearities matter. Our hypothesis is that these details represent patterns that appear repeatedly in the target function. This seems particularly true for the algorithmic tasks (where we see the biggest gains) since the target function is essentially a periodic repetition of peculiar patterns (with sharp decision boundaries). ReLUs can represent such repeating patterns (given enough capacity), but imprinting them *once in the activation function* is a more efficient encoding. This is admittedly a bit handwavy, and needs work to be turned into testable predictions.
>
> [7] *Sampling Theory Perspective on Activations for Implicit Neural Representations*, Saratchandran et al., 2024
>
> [8] *WIRE: Wavelet Implicit Neural Representations*, Saragadam et al., 2023
>
> [9] *Neural Redshift: Random Networks are not Random Functions*, Teney et al., 2024
>
> [10] *Is SGD a Bayesian sampler? Well, almost*, Mingard et al., 2020

---

### Official Review · Reviewer_ZLpT · 2025-11-03

**Soundness:** 3
**Presentation:** 3
**Contribution:** 2
**Rating:** 4
**Confidence:** 4

**Summary:**

The authors propose to parameterize the activation functions of transformers (including MLPs, Attention, and GLU) using splines and analyze how changing the inductive biases (by parameterizing the activation functions) of transformers can influence the final performance of the model along with its training stability and convergence time. They observe that in algorithmic tasks parametrizing the activation functions help in faster convergence along with improved training stability. However, the parameterized activation functions are very task specific and do not generalize well across other tasks unless they are very similar. They further scale their method on larger datasets including tiny stories and observe that they achieve improved performance over using the standard activation functions like Relu and Gelu. However, in this case the improvements are smaller but the transferability is also better. In summary, this work tries to analyze how modifying the inductive biases (via changing the activation functions) can influence the learning process in transformers.

**Strengths:**

* Developing a better understanding about the inductive biases in neural networks is very important in order to understand generalization. And authors investigate an important problem by analyzing the inductive biases induced due to network activations. In this regard, I really liked the approach adopted by the authors to parameterize the activation functions with splines.
* The empirical results on algorithmic tasks are interesting and clearly demonstrate the importance of inductive biases induced by the activation functions.
* Parameterizing the activation functions using splines is interesting because it provides flexibility to the model to adapt as per the task complexity. Intuitively, a simple task would learn simpler activation functions and the complex ones would learn complex activation functions. Therefore parameterizing the activations (if done properly) should help the model adapt to the desired complexity. This would help in improved convergence and performance as shown by this work on simple tasks.

**Weaknesses:**

* Many of the contributions in this paper are quite similar to [1]. For instance, [1] also shows that neural networks when trained on modulo addition converge faster as shown by this paper. Similarly, [1] also parameterizes activation functions using b-splines as done in this work. It would be very helpful if the authors can share more about how their work is different. If the authors are trying to demonstrate that the main differentiating part is to show the effectiveness of parameterizing activation functions on transformers, then it would be very helpful if more evidence on larger models (upto 6-10 layers) and multiple heads is shown. One of my concerns is that scaling this method might lead to overfitting to specific tasks, and thus the generalizability of this method wont be observed.
* Although the current experiments are interesting, I think more evidence demonstrating the importance of parameterizing the activation functions is needed for larger models. It would be helpful if the authors can scale up their experiments. Currently most of the experiments are limited to using relu and gelu activation functions, but in practice we use a few others also like tanh, swish. It would be interesting to have them as baselines. I think the paper would benefit from having evaluations on different architecture choices like rnns, mamba, etc.
* As a side note, at some places the authors should try to tone down their claims, if possible. For instance, in lines 430-431, demonstrating their method being close to the activation functions used in practice, doesn't necessarily mean that current architectures are close to optimal ones.

[1] Do We Always Need the Simplicity Bias? Looking for Optimal Inductive Biases in the Wild (https://arxiv.org/abs/2503.10065)

**Questions:**

* It is not completely clear on how the activation functions should be parameterized. Why is using splines one of the best choices? And if possible, can the authors share more details on how they choose the basis function of splines? My understanding is that they are using linear basis functions? Is there any reason for this? And why did the authors decide to choose 122 points equally spaced between [-20,20]? Is there some reason to come up with these design choices? Just out of curiosity, is it possible to parameterize the spline spacing hyper-parameter?
* An alternate way to learn the activation functions is to use the method proposed in this work [1]. Is there any reason why authors did not use this method? Does it not work in their setup?
* Did authors observe any overfitting? I would expect that parameterizing the activation functions would make the models more vulnerable to overfitting the training dataset and thus hamper generalization. This overfitting could become worse for larger models.
* Simplicity bias says that neural networks prefer learning simpler functions. However, in most cases the activation function learned by authors seems to have a large complexity (at least in fourrier basis). Even though a simpler explanation (say by learning relu) could give high likelihood on training dataset, the authors observe that simpler functions are not learned. Is it showing that simplicity is not always preferred? Or is it that with more data points, the activation functions will asymptotically converge towards relu or a simpler function? Is there some explanation that can justify this?

[1] Do We Always Need the Simplicity Bias? Looking for Optimal Inductive Biases in the Wild (https://arxiv.org/abs/2503.10065)

---

> ### Author Response · Authors · 2025-11-13
> **Response from the Authors**
>
> Thanks for the thoughtful review. We provide clarifications below that we are also adding to the paper.
>
> ----
>
> **W1: Connection with [1]**
>
> This work indeed follows up [1] which left many outstanding questions because it was limited to tiny MLPs/toy data and relied on a bi-level optimization unsuitable to modern architectures. Key innovations:
> - formulation for transformers relevant both to MLPs and attention layers, compatible with modern machinery (normalizations, SwiGLUs) and a tractable optimization method;
> - study of mainstream domains (language modeling, algorithmic reasoning);
> - study of cross-task compatibility ([1] focuses on individual datasets);
> - demonstration of massive improvements on algorithmic tasks (for what they're worth!);
> - a PyTorch implementation that allows swapping activations for ours in any transformer (to reproduce "phase 2" with a different dataset, scale, or architecture) in a few lines of code ([1] did not release any code).
>
> ---
>
> **W2: Larger scale**
>
> Fair request! We have been re-running language modeling experiments:
> - on a range of scales in width, depth. and number of heads (currently up to 12 layers):
> - on top of a very strong baseline (NanoGPT speedrun with rotary embeddings, QK-norm, Muon optimizer, sliding-window attention, mixed-precision training, etc.).
>
> The results so far align with the small scale, including a slight improvement in convergence in number of steps (on FineWeb) for a similar wall-clock time (**edit: see new results in the revised PDF, Appendix E-F**). We are not claiming direct relevance to industrial LLM training, but these results support the paper's findings about the alignment/compatibility of inductive biases.
>
> Other baselines (GeLU, gated units, Swish, TanH, ...) are in Table 3. RNNs and Mamba are outside the scope of this paper.
>
> ---
>
> **Q1: spline parameterization**
>
> This should indeed have been clearer; we are updating the manuscript accordingly. In short, linear splines offer expressivity, unbiased prior, and low cost.
>
> The goal is to learn functions free of priors such as smoothness or monotonicity (unlike e.g. [2,3,4]). Existing methods that use 1-dimensional MLPs for example inherit a smoothness prior and cannot capture sharp transitions that can be useful [1]. Moreover, given fine-enough resolution, linear and high-order splines (e.g. cubic) behave nearly identically (see [1, Appendix D], confirmed in our experiments).
>
> We pick the support and resolution based on the typical range of magnitudes in trained models. Larger values are computationally wasteful. Regarding the optimization of knot placement, this would likely get stuck in local optima.
>
> [2] *Adaptive parametric activation*, Alexandridis et al. 2024
>
> [3] *A survey on modern trainable activation functions*, Apicella et al. 2021
>
> [4] *Kafnets: Kernel-based nonparametric activation  functions for neural networks*, Scardapane et al. 2019.
>
> ---
>
> **Q2: why not use the method from [1]?**
>
> Computational cost. Their meta-learning/bi-level approach is unfeasible with modern architectures. We use an approximation akin to first-order approximations [5] commonly used in meta learning [6].
>
> [5] *Reptile: a scalable metalearning algorithm*, Nichol et al. 2018
>
> [6] *Model-agnostic meta-learning for fast adaptation of deep networks*, Finn et al. 2017
>
> ---
>
> **Q3: overfitting**
>
> There are two possible forms of overfitting. We did not observe either.
>
> - Traditional overfitting to the training data. Does not happen because the activation function has low capacity (~120 scalars), with no room to memorize training data. The evaluation could catch it since we perform phases I and II with disjoint data splits.
>
> - Overfitting of the activation to specific initial random weights. This would mean that the weights and activation function co-adapt, hence the activation function would then capture something about a specific optimization trajectory rather than about the inductive biases necessary for the given task (which are the topic of our study).
> Our method for multi-model training is specifically designed to avoid this. The evaluation would catch this undesirable behaviour because phases I and II use a different random seed.
>
> ---
>
> **Q4: learned activation function have a large complexity (...) simplicity is not always preferred?**
>
> Yes, this is exactly right and the conclusion of [1] (extreme simplicity not always preferrable). A major contribution here is to support this statement for mainstream tasks and architectures (language modeling, transformers). Intuitively, ReLU networks can represent complex functions given enough capacity. But if the target consistently requires complex patterns (e.g. sharp boundaries), these can be encoded more efficiently in activation functions, making the weights' optimization and capacity less of a limiter. This is a key practical implication of our study.

---

### Author Response · Authors · 2025-11-23
**Revision uploaded**

We have uploaded a revision with the following new results and clarifications (modified text in blue in the PDF).

- New paragraph: rationale for splines (Section 2).
- New paragraph: related work on learnability in linguistics and cognitive science (Appendix A), including additional references suggested by reviewers zxSg and 2nXb.
- New paragraph: clarification of the differences with prior work (Appendix A).
- Revised discussion about the results on language modeling (Section 4.1)
- New section: new results with larger models (Appendix E).
- New section: efficient implementation of learned splines, incl. new experiments (Appendix F).
- Mention of future work on ViTs (Section 6).

**Edit (Nov. 28):** we updated Table 6 in Appendix E with more baselines (linear, ReLU, GeLU) and with results across 3 random seeds. The improvements of the optimized non-linearities are now clearer and consistent across tested scales up to 12 layers.


----



**Summary for the AC**
--

**Reviewer ZLpT** (rating 4) noted that "*the authors investigate an important problem (...) I really liked the approach adopted (...)*". They asked:
- for a clarification of differences with prior work by *Teney et al.*: this was indeed not clear enough and we updated the paper on the many differences in the methods, aims, and findings;
- for larger scale experiments: we performed new larger-scale experiments with  and added two new sections (Appendix E-F);
- a few questions on the interpretation of the results: we added technical details and intutiive explanations to the paper.

**Reviewer 2nXb** (rating 8) called this a "*strong and interesting paper, and any weaknesses are relatively minor*".
They provided useful suggestions, we added e.g. a discussion on learnability in linguistics to the related work.

**Reviewer A5zP** (rating 6) noted that "*this work addresses an interesting and important outstanding question*" and that "*the results regarding the effect of scale are interesting and situate these results with the field*".
They provided interesting observations that led us to improve the discussion of our results.

**Reviewer zxSg** (rating 4) made general/basic comments (listed under *Weaknesses*) that seem like remarks rather than deficiencies of the paper. We clarified the fact that this work is about a fundamental scientific question in ML (*Are inductive biases compatible across tasks?*) rather than a method ready to train better LLMs. This should help set up the right expectations. We also used the provided comments to add clarification to the paper that should be helpful to future readers.

Overall, the reviews were encouraging and helped improve the quality of the paper.

**Thanks a lot to the reviewers and ACs for their time and effort.**

---

### Comment · Area_Chair_7aog · 2025-11-28

Dear Reviewers,

The discussion phase is now underway, and the authors have finished uploading their responses to reviewers. If you haven't already, please carefully review the authors' responses to understand their perspectives. Engage in thoughtful, constructive discussions with authors, sharing your thoughts and seeking clarifications. Please also update your review or rating if necessary.

It is noted in the guideline that reviewers can leave comments visible to authors **until Dec 2 11:59pm AoE**. Your active participation and contribution to the ongoing discussion are highly encouraged. Thank you very much for your contribution to ICLR.

Best regards,

AC

---

### Meta-Review · Area_Chair_rjHg · 2026-01-02

**Summary:**

This paper brings new and interesting findings about simplicity in modern networks and on complex tasks. My take away from the reviews and the rebuttals is that there is good reason to accept this paper. Most of the reviewer concerns are addressed in the latest draft and the rebuttals are convincing from my perspective. In particular, this paper has four reviewers whose initial scores average to 5.5 and the resulting changes to the draft improve the paper in my opinion.

**Reviewer Concerns:**

Reviewer ZLpT was mostly concerned with the connection and possible overlap with work by Teney et al., 2025 and also with experiment scale. I read the justification for following up on prior work and on the newer results in the draft and I feel these initial concerns have been addressed adequately.

Reviewer 2nXb advocated for a high score from the first review. I see nothing in their review that looks problematic.

Reviewer A5zP had minor concerns about clarity and claim scope that the authors addressed.

Reviewer zxSg gave an initial score of 4, but the rebuttal responds well and makes a case I agree with that the weaknesses the reviewer lists are more remarks than issues with accuracy, scope, scale, or reproducibility. Thus this score does not keep me from recommending we accept the paper.

**Reviewer Scores:**

Reviewer ZLpT started with a score of 4, and I think upon careful review of the rebuttal and updated draft, would likely have moved above the accept threshold.
Reviewer 2nXb started with a score of 8, I see no reason for this score to have moved.
Reviewer A5zP started with a score of 6 and I would expect it might stay the same or improve slightly.
Reviewer zxSg started with a score of 4, but since I don't see the compelling evidence in the review to arrive at 4, I have a hard time predicting their update here.

(I understand the circumstances of this review process and I am participating as an emergency AC. I am answering all questions asked of me. Please note that this type of prediction of what some other person might do in the face of a rebuttal seems complex and fuzzy and prone to human error. Perhaps it's the best way to analyze the input from the reviewers who spent their time and the authors who invested their time, but I realize I am likely wrong on several predictions of other peoples thoughts/actions.)

---

### Decision · Program_Chairs · 2026-01-26

Accept (Poster)